# Deformation and seismicity decline before the 2021 Fagradalsfjall eruption

Freysteinn Sigmundsson[1 ✉], Michelle Parks[2], Andrew Hooper[3], Halldór Geirsson[1], Kristín S. Vogfjörd[2], Vincent Drouin[2], Benedikt G. Ófeigsson[2], Sigrún Hreinsdóttir[4], Sigurlaug Hjaltadóttir[2], Kristín Jónsdóttir[2], Páll Einarsson[1], Sara Barsotti[2], Josef Horálek[5] & Thorbjörg Ágústsdóttir[6]

Increased rates of deformation and seismicity are well-established precursors to volcanic eruptions, and their interpretation forms the basis for eruption warnings worldwide. Rates of ground displacement and the number of earthquakes escalate before many eruptions[1–3], as magma forces its way towards the surface. However, the pre-eruptive patterns of deformation and seismicity vary widely. Here we show how an eruption beginning on 19 March 2021 at Fagradalsfjall, Iceland, was preceded by a period of tectonic stress release ending with a decline in deformation and seismicity over several days preceding the eruption onset. High rates of deformation and seismicity occurred from 24 February to mid-March in relation to gradual emplacement of an approximately 9-km-long magma-filled dyke, between the surface and 8 km depth (volume approximately $34 \times 10^6$ m³), as well as the triggering of strike-slip earthquakes up to magnitude $M_W$ 5.64. As stored tectonic stress was systematically released, there was less lateral migration of magma and a reduction in both the deformation rates and seismicity. Weaker crust near the surface may also have contributed to reduced seismicity, as the depth of active magma emplacement progressively shallowed. This demonstrates that the interaction between volcanoes and tectonic stress as well as crustal layering need to be fully considered when forecasting eruptions.

Volcano observatories worldwide aim to provide timely eruption warnings to civilians, aviation authorities and other stakeholders, to prevent loss of life and damage of infrastructure. To achieve this, it is essential to correctly understand the pattern of eruption precursors[4–6], which often show escalating rates before eruption onset[2,3]. A pioneering work to quantify such behaviour was that by Voight[1], who derived a method of interpreting escalating rates of observed precursory activity at volcanoes such as strain or seismicity, to find the time of failure and eruption onset. This material failure forecast method, or modified versions of it, have been widely used to anticipate the timing of eruptions in hindsight, and in some cases in near-real time[7–9]. However, some eruptions show a different behaviour, even a reduction in precursory activity immediately before eruption. The flank eruption of Eyjafjallajökull volcano in Iceland in March 2010, which occurred before the explosive summit eruption, was preceded by a decline in seismic activity and deformation rates after several months of elevated activity[10], with no immediate short-term warning issued before it began[11]. A reduction in seismicity has also been noted at some stratovolcanoes before phreatic explosions or eruption onset, for example, before the activity in 1989–1990 and 2009 at Redoubt[12,13] and before that at Telica in 1999 (ref. [14]). In some cases, this has been attributed to sealing of gas migration pathways, which in turn leads to pressurization of the system[15] and

a resultant increase in surface deformation (uplift)[16]. The precursory activity we observe at Fagradalsfjall, however, involves both a decline in seismicity and deformation.

## Pre-eruptive seismicity and deformation

The Reykjanes Peninsula oblique rift zone is a part of the North American–Eurasian divergent plate boundary where it emerges above sea level in Iceland (Fig. 1 and Extended Data Fig. 1). Plate spreading of 19 mm yr⁻¹ in a direction of around N104°E is highly oblique to the central axis of the Reykjanes Peninsula, which is directed at around N77°E, with a large component of shearing across the zone compared to opening[17,18]. En echelon arranged fissure swarms aligned north east to south west along the Reykjanes Peninsula have formed the basis for dividing the zone into volcanic systems (comparable to spreading centres on mid-ocean ridges), each consisting of a fissure swarm and a high-temperature geothermal area[19]. Arrays of strike-slip faults with north–south orientation have been mapped, mostly between the fissure swarms, releasing shear stress across the plate boundary during non-eruptive periods[20]. Before the eruption that began on 19 March 2021 in Geldingadalir at Mount Fagradalsfjall, no eruption had occurred on the Reykjanes Peninsula for around 800 years. Volcanic activity in the last 3,000 years is characterized

[1]Nordic Volcanological Center, Institute of Earth Sciences, University of Iceland, Reykjavik, Iceland. [2]Icelandic Meteorological Office, Reykjavik, Iceland. [3]Centre for the Observation and Modelling of Earthquakes and Tectonics (COMET), School of Earth and Environment, University of Leeds, Leeds, UK. [4]GNS Science, Lower Hutt, New Zealand. [5]Institute of Geophysics, Czech Academy of Sciences, Prague 4, Prague, Czech Republic. [6]Iceland GeoSurvey, ÍSOR, Urðarhvarf 8, Kopavogur, Iceland. ✉e-mail: fs@hi.is

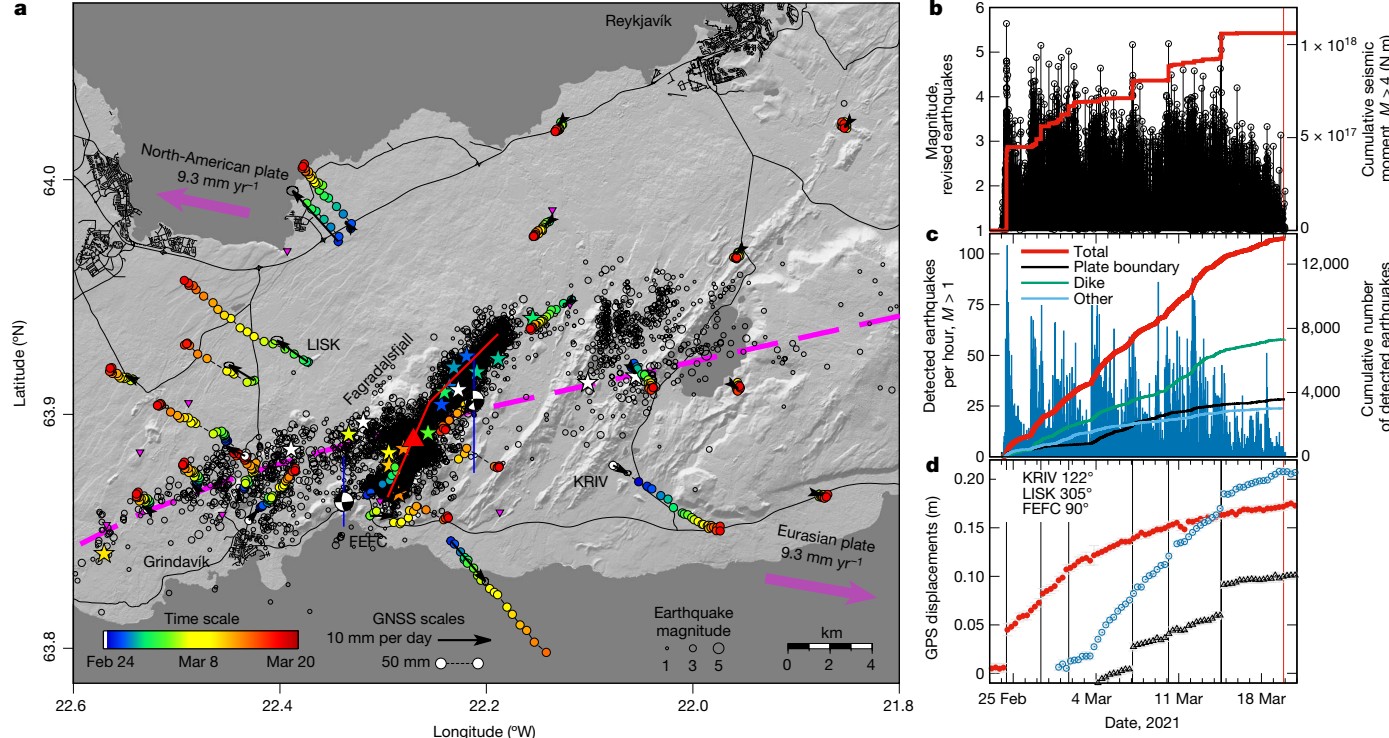

**Fig. 1 | Overview of seismicity and deformation. a**, Map of the Reykjanes Peninsula. Manually reviewed earthquake locations $M > 1$, covering the period from 24 February to 19 March 2021, are shown as black open circles, with earthquakes $M_W > 4.5$ marked with stars (the colour scale represents time evolution, see legend). Seismic stations are marked with magenta inverted triangles. Focal mechanisms of the two largest earthquakes ($M_W$ 5.64 on 24 February; $M_W$ 5.3 on 14 March) are shown, as well as representative north–south fault locations (blue lines). Daily cumulative horizontal displacements at GNSS stations are shown with coloured circles and mean velocities during the time period as black arrows. The magenta dashed line shows the approximate plate boundary central axis and the magenta arrows show relative plate motion vectors[17]. The eruption site is marked with a red triangle and the approximate surface projection of the dyke with a red line. **b**, Magnitudes (black) and cumulative seismic moment (red) based on manually revised earthquakes with $M > 1$. **c**, Hourly earthquake rate (blue) and cumulative number of earthquakes (red) in the whole study area based on automatically located earthquakes with $M > 1$. Also shown is the cumulative number of earthquakes divided into subareas (Methods). **d**, Rotated horizontal (maximum) displacements from three GNSS stations (see map for locations). Displacements at KRIV are displayed in red, at LISK in blue and at FEFC in black. Eruption onset (20:40 on 19 March) and largest earthquakes shown with vertical lines.

by eruptive periods of a few hundred years, separated by 800–1,000 years of no eruptive activity[19,21]. Geodetic measurements from the 1990s have shown strain and stress accumulation along the plate boundary, in agreement with plate motion models and a locking depth of about 5–8 km, below which ductile deformation dominates and the tectonic plates slide freely, perturbed by earthquake and geothermal deformation[22–24] (Extended Data Fig. 1). Unrest began on the Reykjanes Peninsula in mid-December 2019 with an intense, week-long earthquake swarm at the southern margin of Mount Fagradalsfjall, followed by elevated seismicity along a large part of the Reykjanes Peninsula. Periods of high-intensity swarms related to repeated magmatic intrusions, seismic triggering and tectonic activity occurred in different areas, with several episodes of inflation identified in three areas along the plate boundary[25,26]. Here we describe the activity from 24 February 2021, when the largest earthquake in the unrest episode occurred (magnitude $M_W$ 5.64), until the onset of the eruption on 19 March 2021.

The $M_W$ 5.64 earthquake on 24 February 2021 was preceded by 3 h of intense, concentrated microearthquake activity, at the forthcoming dyke location, with several events per minute occurring in a narrow depth range near 7 km depth, suggesting the start of a magmatic intrusion. After propagation several hundred metres northeast along the approximate direction of the future dyke, the earthquake swarm culminated in a $M4$ strike-slip event, triggered within the swarm area. Within 17 s the $M_W$ 5.64 earthquake was triggered 1 km to the southeast, near the central axis of the plate boundary (Fig. 1a). During the next 4 h all activity was confined to the central axis of the plate boundary where

10 km long segments to either side of the earthquake were activated with over 60, mostly strike-slip, $M > 3$ earthquakes. Within the first 2.5 h, eight $M_W > 4.0$ events occurred along these segments indicating slip on around 20 km of the plate boundary (Fig. 1a and Extended Data Fig. 2a). The deformation pattern in the area changed after 24 February, suggesting sustained inflow of magma into a vertical dyke in the brittle crust and some slip on the central axis of the plate boundary. Ground deformation indicative of dyke emplacement is well documented by Global Navigation and Satellite System (GNSS) geodesy and interferometric analysis of synthetic aperture radar images (InSAR). KRIV GNSS station to the south east of the dyke displays a co-seismic jump during the $M_W$ 5.64 earthquake and then displacement at a rate of approximately 10 mm per day in a direction around N148°E over the following days, gradually declining with time (Fig. 1b). LISK GNSS station to the north west of the dyke, moved at a rate of around 4 mm per day in a direction N305°E, until 3 March, when the deformation sped up to around 14 mm per day in a similar direction, but thereafter decreasing gradually with time. Interferograms spanning 6 days, formed from Sentinel-1 data, reveal how displacements in the line of sight (LOS), from the satellite to ground, change over time in February and March 2021. Also evident from InSAR is a decrease in deformation rate over time, with the slowest deformation rate occurring in the last few days before the eruption (Fig. 2 and Extended Data Fig. 2). Both the GNSS and InSAR data thus reveal a high rate of deformation after the $M_W$ 5.64 earthquake, which then gradually decreases to almost zero at the eruption onset.

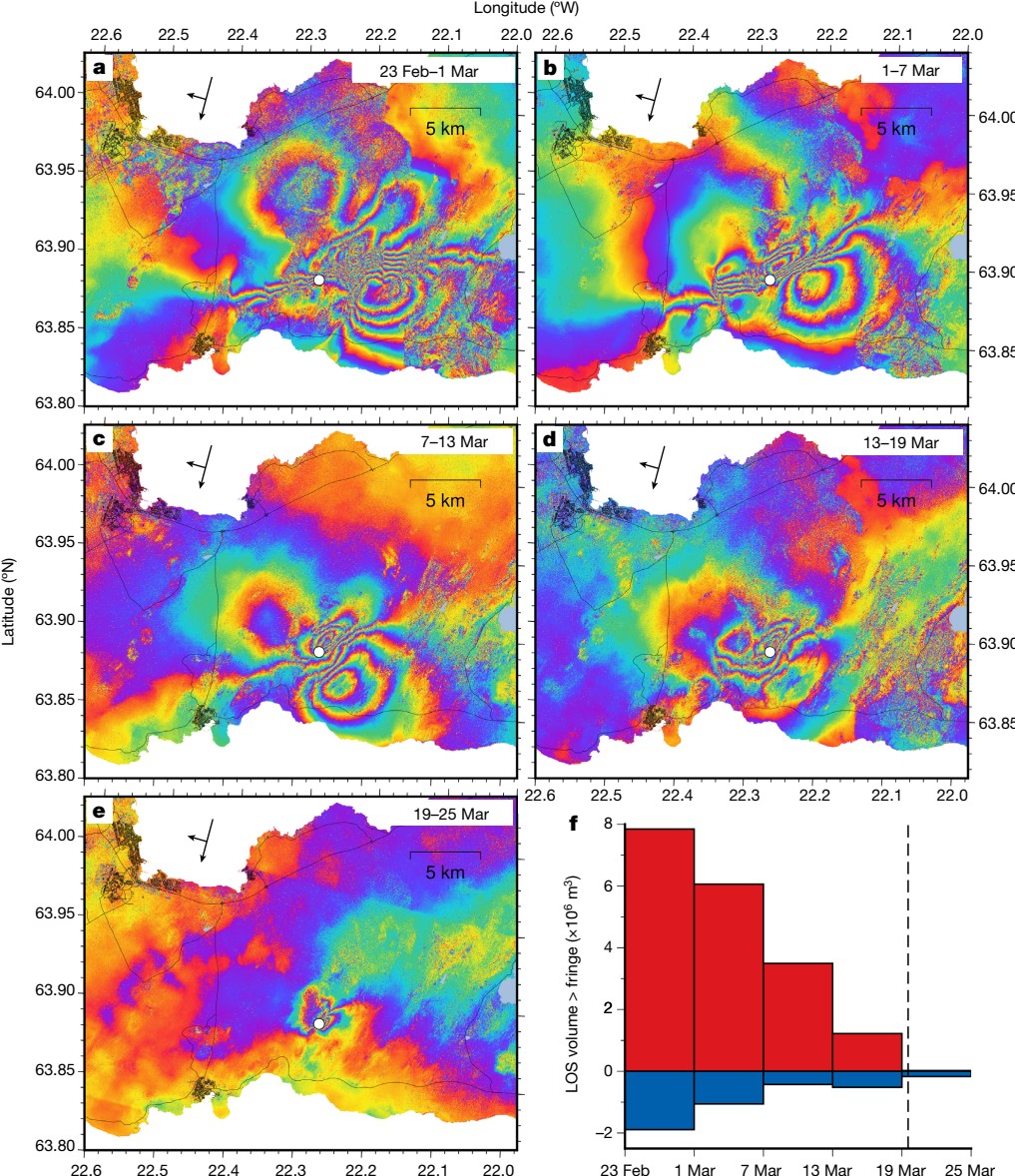

**Fig. 2 | Spatial evolution of deformation. a–e**, InSAR interferograms from Sentinel-1A and Sentinel-1B images from descending track T155 (acquired 07:58 local time), spanning 6 days each, showing LOS change in wrapped phase (one full fringe pattern corresponding to 28 mm phase change). Black arrows show the satellite flight and viewing direction. The white circle shows the eruption site on 19 March 2021. **f**, Red, positive LOS volume (towards the satellite) for each inteferogram defined as the sum of unwrapped LOS values in all pixels passing a threshold value of 28 mm (one fringe) multiplied by the pixel area. Blue, same for negative LOS volume, negative values exceeding −28 mm at individual pixels multiplied by pixel area.

Following the $M_W$ 5.64 earthquake on 24 February 2021, nine earthquakes greater than $M_W$ 4.8 occurred on the Reykjanes Peninsula before the eruption, with the largest of these being a $M_W$ 5.33 event on 14 March 2021 (Methods and Supplementary Table 1). The earthquake magnitudes and rate of seismic moment release decreased until the eruption onset in the evening of 19 March (Fig. 1b), with only one event above $M_W$ 4.0 ($M_W$ 4.20 on 15 March). Two days before the eruption, the seismicity was mainly concentrated in two swarms of small earthquakes, one of them at the forthcoming eruption site. From 24 February until the eruption onset on 19 March over 53,000 earthquakes were automatically detected (Methods). A characteristic of this seismic activity is an unusually large number (64) of $M_W$ ≥ 4.0 earthquakes, nearly 20 more than recorded in the area during the whole two preceding decades. The unprecedented earthquake activity, characterized by overlapping mainshock–aftershock sequences, was felt day and night for three weeks in the capital area of Iceland, as north–south

oriented faults were triggered along the peninsula when Coulomb failure stress increased by up to 4 MPa in response to the opening along the dyke (Extended Data Fig. 1). Relative relocations of seismicity at the dyke, obtained through cross-correlation of waveforms (Methods), reveal a vertical two-segment dyke, with a northern segment striking N45°E and a southern segment N25°E (Extended Data Fig. 3). Seismicity at the dyke occurred along the northern segment from 24 February to 3 March. Thereafter, seismicity migrated towards the dyke's southern segment and also westwards, activating an approximately 15 km segment of the plate boundary over the next 4 days. On 7 March, this trend ceased and activity migrated from the dyke centre to the southwest along the southern dyke segment. Seismicity rate, influenced by the overall evolution of both the dyke and plate boundary activity, shows an overall decline in the few days before eruption onset. The decline begins 3 days earlier in the plate boundary area compared to the dyke area (Fig. 1c). In the last 15 h before the eruption, seismic signals depleted

in higher frequencies, characteristic of shallow sources, were recorded from about 20 shallow (<1 km depth), less than *M*2 earthquakes at the forthcoming eruption site.

## Geodetic modelling

The sources of surface deformation were inferred from modelling using a modified version of the GBIS geodetic Bayesian inversion software[27]. Co-seismic deformation was modelled as a result of shear movement across rectangular planes, and a dyke with opening and shearing on planes, embedded within a uniform elastic halfspace (Methods). The observed surface deformation, also analogous to the seismicity, reveals activity along the central axis of the plate boundary during the study period. Similar, but smaller, deformation was also present in the years preceding the unrest (Extended Data Fig. 4) and can be explained with a model of shearing on a subvertical plane trending along the plate boundary, which we also include in our model. Shearing along this plane and the dyke plane combine to reproduce effects of distributed shearing in the area that may have occurred on a series of closely spaced north–south oriented strike slip faults mapped in the area[20,21]. A localized zone of subsidence near the dyke centre is additionally modelled here as a result of a gradually contracting point source of pressure throughout the events. The modelling shows that the majority of the deformation relates to emplacement of a vertical dyke in the area where a large part of the smaller-sized earthquakes occurred. An initial model considers a one-segment dyke with uniform opening and shearing (Extended Data Fig. 5). This model is then improved by considering a two-segment dyke, in line with relocated seismicity, with distributed opening and slip (Methods and Extended Data Fig. 6). The northern segment strikes N45°E and the southern segment N23.5°E (Fig. 3a). Daily volume change (magma inflow rate) is solved for (Methods) and the corresponding mean depth of magma emplacement is calculated (Fig. 3b, c). These are constrained overall by the joint GNSS and InSAR data, with GNSS data constraining the daily change. The inferred magma inflow rate (volume change) into the dyke fluctuates. Highest values occur in the initial days of the dyke formation from 24 February to 3 March (30–35 m³ s⁻¹), values in the range of 10–20 m³ s⁻¹ are inferred during 3–15 March and the lowest values (<10 m³ s⁻¹) occur in the days before the eruption onset (15–19 March). The daily mean depth of magma emplacement also fluctuates, but after 11 March it becomes progressively shallower. There appears to be a linear relationship between magma flow rate and mean depth of emplacement, so the shallower the magma emplacement, the lower the magma flow rate (Fig. 3c). This is consistent with the flow rate being controlled by the flow up a conduit below the dyke with no significant pressure required to intrude along the dyke (Methods). The correlation predicts an initial eruption rate of around 7 m³ s⁻¹, in agreement with observations[28]. It also allows an estimation of the depth of the source feeding the dyke at 19 km, in agreement with geochemical observations[29]. The inferred cross-sectional area of the conduit, connecting the magma source to the bottom of the dyke, is of the order of a few square metres.

The combined effect of the modelled deformation sources (Fig. 4) is to release stress accumulated by plate movements. The estimated dilatational stress change at 3.5 km depth in the model is tens of megapascals (Extended Data Fig. 7), whereas the previous yearly dilatational stress change due to plate movement is three orders of magnitude smaller or tens of kilopascals per year (Extended Data Fig. 1). The stress change during the events may therefore correspond to a significant or large fraction of tectonic stresses accumulated over time since the previous eruptions on the Reykjanes Peninsula, about 800 years ago. The derivative of east displacement in the north direction for the full period of the dyke intrusion, which can be compared directly to InSAR observations (Fig. 4b,c), illustrates well how the central axis of the plate boundary also plays an important role in releasing shear stress. This shearing together with the dyke opening combine to release tectonic stress in this oblique spreading setting (see also Extended Data Fig. 7). The spatial pattern

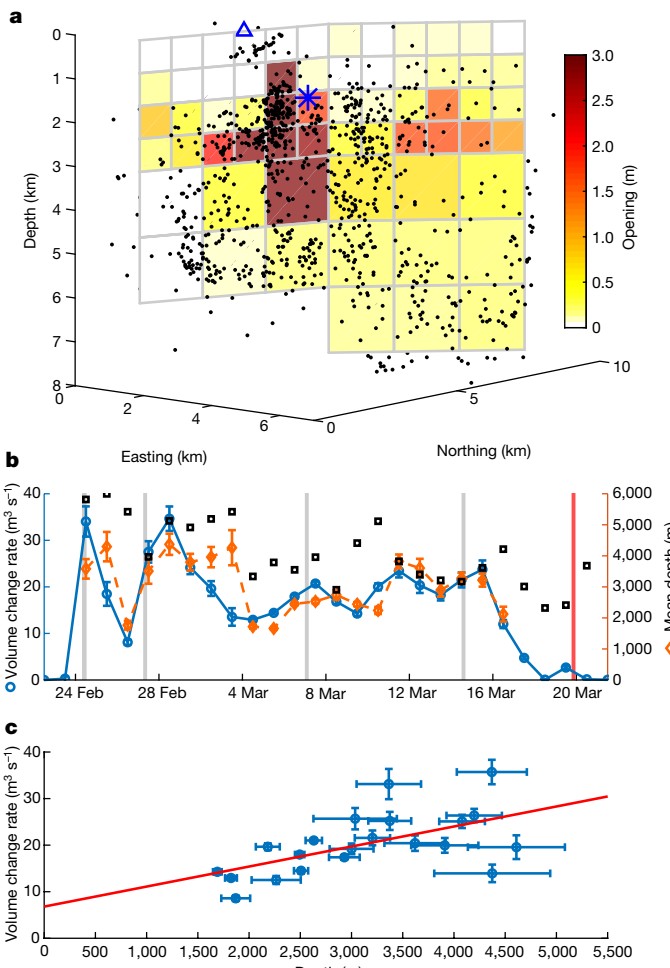

**Fig. 3 | Geodetic modelling results. a**, Opening along the dyke from 23 February to 12 March 2021. Colour represents the median value for each patch from posterior probability distributions. The blue triangle denotes the eruption site and the blue star is the additonal contracting point source of pressure. **b**, Daily volume change rate and mean depth of the volume change (Methods). Depth can only be reliably estimated when the volume change is significant and we use a threshold of 5 m³ s⁻¹. Error bars represent 1 standard deviation. Grey vertical bars indicate the times of the largest earthquakes (*M*_W > 5) and the red bar shows the eruption onset. Black squares indicate the daily mean earthquake depth. **c**, Values from **b** plotted against each other. The red line is the best fitting line using the least squares method.

of the surface displacement due to the cumulative earthquake slip, excluding the two largest events, is of a similar form to that due to the combined effects of the other sources (Extended Data Fig. 9), showing that the smaller earthquakes also contribute to stress release.

## Implications

Our observations suggest that a release of tectonic stress followed by a decline in deformation and seismicity rate may be a characteristic precursory activity anticipated for a certain class of eruptions. Our observations are consistent with the following progressive stages: (1) Magma flows into a dyke with associated characteristic deformation and seismicity, as well as triggered deformation and seismic activity in surrounding areas. Tectonic stress accumulated before the activity is relieved during a period of high strain and stress release. (2) The main period of tectonic stress release is followed by declining deformation

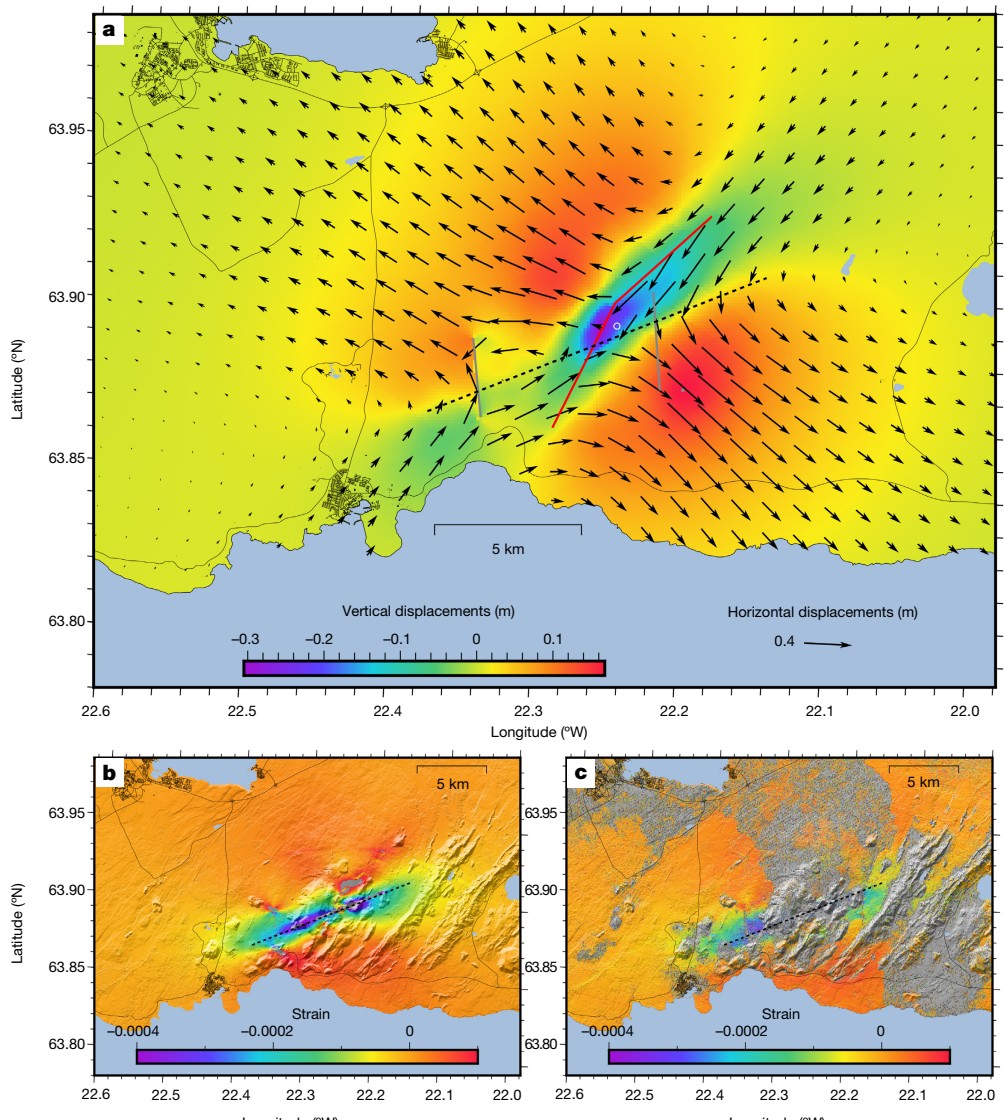

**Fig. 4 | Modelled ground displacement and simple shear. a**, Modelled cumulative surface deformation due to all sources. Sources are shown as follows: dyke (red line), plate boundary (dashed line), two largest earthquake faults (grey lines), and deflation source (white circle). **b**, The derivative of east displacement in the northerly direction (simple shear) from the model of the full period of the dyke intrusion. This shear occurs at the modelled plate boundary segment. **c**, Same as **b** but observed from a combination of ascending and descending satellite tracks for the period of the dyke intrusion[41].

and seismicity rates, both associated with a decline in magma inflow rate. As stress is gradually released at the bottom of the brittle crust, the magma is forced to travel higher before intruding along the dyke. This leads to a greater drop in driving pressure, and hence a decrease in magma flow rate. (3) Once accumulated tectonic stress is released along the dyke, magma breaches the surface in a relatively calm manner, initiating an eruption without significant seismic energy release or deformation. Even if the dyke needs to break through the uppermost crust to initiate an eruption, the topmost 1 km of crust in volcanic rifts is fractured and weak so signals are not easily detectable. An eruption may occur without significant further precursory activity. This is a very different pattern of precursory activity from the case where activity escalates before eruptions as anticipated with the material failure forecast method. Although this method predicts escalating rates of precursory activity, it is known that the inflation rate and the rate of magma inflow into a shallow magma body may slow down before eruptions as was the case for Krafla volcano, North Iceland[30,31]. An exponential decline in magma flow rate is expected in a magma conduit

linking a deep source of constant pressure and a shallow magma body where pressure builds up, as the magma flow rate is proportional to the pressure gradient in the channel[32]. If magma is less dense than the surrounding host rock, then magma buoyancy has an important role in driving magma upwards in such channels[32].

Our findings have similarities with observations of passive dyke intrusions and rifting, for example, at Kilauea volcano, Hawaii, in areas of previous high stress and strain accumulation due the sliding south flank of Kilauea[33,34]. Before the September 1999 intrusion, no precursory inflation was observed and no increase in subsurface magma flow rate, indicating no premonitory increase in pressure within the magma plumbing system. In East Africa, precursors to passive rifting[35–37] are influenced by high tensional stress. Precursory activity that we observe also resembles that observed before the 1975–1976 fissure eruption at Tolbachick volcano, Kamchatka, where earthquake swarms sharply decreased 1–2 days before the opening of new fissures[38]. The precursory behaviour we observe may be relevant for eruptions in volcanic rifts in general, that follow the formation of an evolving dyke intrusion fed from a deep magma body into a

pre-stressed crust. In such tectonic settings, the unstressing of the crust should be expected before eruptions. This may, for example, have significant implications for the next eruption at Mauna Loa, Hawaii, where there has been persistent volcanic unrest[39]. If precursory activity will be similar to that at Fagradalsfjall, then escalating deformation and seismicity may not occur before the next eruption. Monitoring and understanding precursors of volcanic eruptions is paramount for anticipating the potential associated hazards and for reducing the potential impact on people and infrastructure[40]. Therefore, identifying less common pre-eruptive trends in geophysical datasets typically acquired by volcano observatories is an essential step in supporting timely decision-making and risk mitigation measures by civil authorities.

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

# Methods

## Routine earthquake locations and seismicity rate

Seismicity in the area is monitored by the Icelandic Meteorological Office (IMO), where earthquakes are recorded on the SIL national seismic network. Arrival times of P and S waves from the earthquakes are automatically detected by the SIL analysis system, which locates the events and assigns a preliminary magnitude. The events are then manually reviewed, revised and updated as required. These routine analyses use one common one-dimensional velocity model (SIL model) for all events in Iceland[42,43]. The total number of earthquakes automatically detected by the SIL system during the period 24 February to 14 March 2021 was about 45,000, with 15% manually reviewed. In the following period, 15–19 March, 19% of the roughly 7,500 automatically located earthquakes have been reviewed. To improve seismic monitoring of the unrest on Reykjanes Peninsula, eight additional stations on the peninsula, owned by the Czech Academy of Sciences (CAS) and operated by Iceland GeoSurvey (ÍSOR), were connected to IMO's monitoring system and are also used in the automatic and reviewed standard locations. Three of these stations (lsf, iss and lag) were added to the system in 2020 and an additional four (lat, ash, moh, faf) were added in early March 2021 (the first three on 5 March, the fourth on 11 March). On 5 March one station (odf) from University of Cambridge was also included in the routine automatic and manual analyses. The recently increased station density on Reykjanes Peninsula has improved the magnitude of completeness ($Mc$) for seismicity in the area, from its previous 0.5 or better estimate for the period 2002 to 2013 (ref. [44]). However, during intense seismic swarms with large earthquakes the automatic location system saturates and background noise greatly increases, resulting in significant decrease in sensitivity. For the intense period analysed here, $Mc$ for Reykjanes Peninsula is estimated to be around $M1$, and thus the seismicity rate analysis is limited to $M > 1$ earthquakes. Seismicity rate (Fig. 1c) is estimated separately for the dyke and the plate boundary areas, by summing up all events in a 2-km-wide strip around the dyke and a 2.5-km-wide strip around the plate boundary axis (earthquakes in the overlap area are divided between the dyke and the plate boundary areas). Figure 1c also reports number of earthquakes in other areas. With only a fraction of the seismicity having been manually reviewed, the seismicity analysis is based on automatic locations.

## Earthquake focal mechanism and magnitude determination

After manual review of the earthquake locations, focal mechanisms and magnitudes are calculated, and the events added to the SIL catalogue. The calculation is based on a grid search through all possible strike, dip and rake angles for which first-motion P-wave polarities match and for which P- and S-wave amplitudes are within acceptable limits of those observed[45]. Magnitude calculations in the SIL analysis system tend to underestimate magnitudes for events of $M_W > 3.5$ (ref. [46]). Better magnitude estimates for these events are automatically calculated by a separate real-time analysis, which continuously evaluates observed peak ground velocity (PGV) in different frequency bands at each seismic station[47]. With these values and automatic locations from the SIL system for events of $M > 3.5$ the moment magnitude, $M_W$, is calculated for each station using a ground-motion prediction equation for velocity[48]. The final $M_W$ estimate for each event is a robust mean of a third of the calculated values around the median, from stations outside the clipping range (around 15 km) to a distance of 230 km from the event[49]. This value is robust and compares well with internationally determined moment magnitudes. Supplementary Table 1 lists all of the determined magnitudes for $M_W \geq 4.0$ events on Reykjanes Peninsula from February to April 2021. These magnitudes are used to calculate the cumulative moment released in the largest events during the unrest period.

## Relative relocation of earthquakes

The seismic activity during the period 24 February to 20 March 2021 is located on and around both the dyke intrusion and the many approximately north–south strike-slip faults triggered by the intrusion. Figure 1 and Supplementary Table 1 show that 15 strike-slip events of $M \geq 4.5$ were triggered near the dyke, resulting in many aftershocks on their fault planes in the magnitude ranges above around $M1.5$. Therefore, in order to map the overall features and dimensions of the dyke intrusion and to minimize the contamination by aftershocks from the approximately north–south strike-slip faults, a data set of smaller magnitude events was selected for high-precision relative relocation. The data set includes all 1,321 earthquakes thus far routinely located and reviewed, in the local magnitude range $0.5 < M < 1.4$.

The relocation method uses double differences of both 'picked' absolute arrival times and improved relative arrival-time measurements obtained through cross-correlation of P and S waveforms from different earthquakes[50]. The procedure iteratively inverts the weighted square sums of absolute P and S arrival-time differences, as well as the double differences of (1) absolute arrival times of P and S waves, (2) relative arrival times of P and S waves and (3) relative S−P arrival times. The resulting distribution of event locations has a high internal/relative location accuracy and is therefore well suited to map details of active subsurface faults and dyke intrusions[51,52]. The velocity structure on the Reykjanes Peninsula is significantly different from the standard SIL-velocity model used in the routine analysis, influencing in particular the hypocentre depths, which tend to be roughly 1 km too deep in the routine analysis. Therefore, a better-fitting one-dimensional velocity model is used, derived from travel-time and phase-velocity analysis of body waves propagating along the Reykjanes Peninsula[53] and the Reykjanes-Iceland Seismic Experiment seismic refraction profile[54]. Fifteen seismic stations within a distance of 50 km from the earthquakes are used in the relocation including all IMO stations and the three CAS/ÍSOR stations connected to the system in 2020. Exclusion of stations closest to the dyke is intentional to fulfil the requirements for the relocation method to properly work, as it assumes that waves from closely located events travel through the same velocity structure. The resulting earthquake hypocentre depths are referenced to the mean elevation of the nearest stations, at approximately 80 m above sea level.

The resulting relocated event distribution (Fig. 3 and Extended Data Fig. 3) shows two main segments in the approximately 9-km-long dyke intrusion, differing in strike and depth range. A northern segment, at least 4.5 km long, has a more easterly strike, of around 45°, and reaches depths of 8 km. A 4.5-km-long southern segment strikes 25° and reaches depths of 6 km. The change in depth is rather sharp at the intersection of the two segments (B on Extended Data Fig. 3). Extended Data Fig. 3 also shows a clustering of events around a depth of 0.5 km in the eruption region (the star and white line on the figure). The colour coding of events in Extended Data Fig. 3 reveals the overall time evolution of the activity during the period. Seismicity begins at a depth near the centre of the dyke (B on Extended Data Fig. 3) and activates the northern segment between 24 February and 3 March, before propagating southwards activating the southern segment from around 4 March to 19 March. The area around the intersection (B on Extended Data Fig. 3) remains active for the whole time period.

## Geodetic modelling

We inverted the geodetic data using a Bayesian approach with a modified version of GBIS software[27] to obtain the range of source parameters that can explain the observed deformation. An initial inversion was carried out using GNSS observations from 18 stations (a combination of campaign and continuous measurements) covering the entire pre-eruptive period from 23 February to 19 March 2021, and two Sentinel-1 interferograms (from descending track T155 covering the period 23 February to 19 March 2021 and ascending track T16 covering

the period 19 February to 21 March 2021). This Bayesian inversion software utilizes a Markov-chain Monte Carlo approach and the Metropolis Hastings algorithm[55,56]. The joint probability distribution function for the various source parameters was obtained by running two million iterations. In our initial inversion we modelled the dyke with a rectangular dislocation[57], allowing for opening and shear, and the observed movement at the central axis of the plate boundary with a second dislocation, also allowing for opening and shear. We modelled the surface displacements resulting from the two largest tectonic earthquakes as separate rectangular dislocations, allowing for shear motion only. A subsidence signal close to the eruption site was modelled as a deflating point source[58] for simplicity, although this deflation may be the result of normal faulting related to the dyke intrusion. We assumed a Poisson's ratio of 0.27. All of the source parameters were allowed to vary, with wide bounds defined a priori.

The output parameters from this inversion confirmed that a dyke was intruding along vertical planes defined by the seismicity. In a second inversion, we substituted the single dyke dislocation for two connected dislocations with strikes fixed to that indicated by the relocated seismicity (N45°E in the northern segment and N23.5°E in the southern segment). We allowed the position to vary, recognizing that the absolute locations for the relocated seismicity are not accurate, and found that the preferred location was around 400 m west of the relocated seismicity and passed through the eruption fissure.

For our final inversion we fixed the location of the dyke to the preferred location from the second inversion with maximum lengths (4.5 km for each segment) and depths (7.5 km in the north and 6 km in the south) derived from the seismicity. We also fixed the location, dip (vertical) and strike (N65.4°E) of the plate boundary dislocation from the second inversion. Very little opening (9 cm) was assigned to this segment in the second inversion, so we allowed for shear motion only. Given the vertical nature of both the dyke and plate boundary segments, we allowed for strike-slip shearing in a strike-slip sense only, with no dip-slip motion. We divided the dyke and plate boundary segment into patches, $750 \times 750$ m$^2$ in the upper 3 km and $1.5 \times 1.5$ km$^2$ below a depth of 3 km, and solved for the opening and/or slip independently on each patch. We allowed the location of the deflating point source to vary, and our model placed it at a depth of 1,480–1,670 m with a volume change in the range of $-2.6$ to $-3.4 \times 10^6$ m$^3$.

To constrain the daily incremental opening along the dyke, we used a constrained linear least squares approach. We used 100 samples from the posterior distribution of our final inversion for the entire pre-eruptive period to constrain the total opening for each dyke patch and solved for incremental opening using GNSS only. To reduce the noise in the GNSS time series, we first filtered them using a 2-day triangular moving window, before calculating the incremental displacements. We estimated the daily opening for each patch, with a constraint that only positive opening was allowed. To account for systematic errors affecting the whole GNSS network, we simultaneously solved for a daily reference frame adjustment in east, north and up. We then summed the daily volume change of every patch to give a daily volume change rate and calculated the mean depth (centre of gravity) of this daily volume change.

The trade-off between daily volume change and mean depth of magma emplacement was studied with a simulation test. An incremental dyke opening inversion was carried out on the simulated data to test whether or not the observed trade-off between depth and volume change for the daily solutions (Fig. 3) influences the overall relationship between depth and volume change. We divided the total opening of the maximum a posteriori probability solution from our distributed opening inversion into 28 days of equal volume change, with shallowing mean depth. We used these daily dyke opening models to simulate the displacements at the GNSS stations and added representative noise. We then inverted the simulated GNSS data in the same way as before. The results are shown in Extended Data Fig. 8. Despite the trade-off between

depth and volume for individual days, the simulated constant volume change rate is retrieved, with no dependence on depth.

Ground deformation due to all 64 recorded earthquakes $M_{\rm w} \geq 4$, except for the two largest events on 24 February and 14 March, was evaluated by the following procedure. Each earthquake was assumed to be on a rectangular north–south striking vertical strike-slip fault in accordance with the dominating fault mechanisms. The central location and depth of each fault was acquired from the IMO hypocentre catalogue. For all earthquakes smaller than $M5$, a rectangular fault of $2 \times 2$ km$^2$ was assumed; for $M \geq 5$ a length of 4 km and a height of 2 km was used for the earthquake. For the mapping of very shallow hypocentres (with depths of less than 1.1 km), 1 km was added to the central depth to avoid artefacts from an artificial fault sticking out of the ground. The moment magnitude was converted to seismic moment $M_0$ using the relation $M_0 = 10^{1.5M+9.05}$ (ref. [59]) and a shear modulus of $\mu = 30$ GPa to obtain the mean slip $s$ from the seismic moment, $M_0 = \mu As$, where $A$ is the fault area. The cumulative moment of all 62 faults (that is, excluding the two largest earthquakes) was equivalent to a single $M_{\rm w}$ 5.9 earthquake. The software Coulomb[60] was used to calculate the surface deformation from the faults in a dense $(0.1 \times 0.1$ km$^2)$ grid. The resulting displacement field (Extended Data Fig. 9) has the same characteristics as the geodetic model produced by the main inversion using distributed slip along the dyke together with the plate boundary segment and the two main faults, but is of much smaller magnitude. Considering the contribution of smaller faults in the inversion process as described above did not improve the fit to the geodetic data, so these are not considered here. The effects of the smaller faults can be expected to be already accounted for in the main inversion.

## Conduit flow modelling

We modeled the flow to the base of the dyke through a cylindrical conduit. In reality, the conduit may be a different shape and more fracture-like when it is forming. However, it is formed in the ductile lower crust below the locking depth of the crust, where plate tectonic stress does not build up in the same manner as in the topmost elastic crust. Furthermore, magma flow in a fracture of limited dimensions may rapidly focus towards the widest part of such a fracture, and the effective flow path may become more cylindrical in shape[32,61,62]. In any case, a non-circular cross section would not significantly affect our conclusions (see below). Laminar flow of an incompressible fluid in the conduit is assumed, driven by the difference between the pressure at the magma source region and the magmastatic head, and by treating the dyke as a reservoir that presents negligible resistance to flow,

$$P - \rho_{\rm m} g(L + h) = \frac{8vL}{r^2} u + \frac{\rho_{\rm m}}{2} u^2, \qquad (1)$$

where $P$ is the pressure at the magma source, $\rho_{\rm m}$ is the density of the magma, $g$ is gravitational acceleration, $L$ and $r$ are the length and radius of the conduit, respectively, $h$ is the height above the base of the dyke at which magma is intruding, $v$ is the dynamic viscosity and $u$ is the mean magma velocity in the conduit. The term on the left is the driving pressure, the first term on the right is the viscous loss due to laminar (Hagen–Poiseuille) flow and the second term on the right is the dynamic pressure loss. Using the relationship between volumetric flow rate and velocity, $Q = \pi r^2 u$,

$$P - \rho_{\rm m} g(L + h) = \frac{8vL}{\pi r^4} Q + \frac{\rho_{\rm m}}{2\pi^2 r^4} Q^2 \approx \frac{8vL}{\pi r^4} Q, \qquad (2)$$

in the case where $L \gg \rho_{\rm m} Q/16\pi v$. Note that in the case when $h = 0$ and $P = \rho_c gh$, where $\rho_c$ is the average density of the crust above the magma source region, this reduces further to the more familiar relationship $Q = \pi r^4 (\rho_c - \rho_{\rm m})g/8v$ (ref. [63]). Differentiating (2) with respect to $h$ gives

$$-\rho_m g = \frac{8\nu L}{\pi r^4}\frac{\mathrm{d}Q}{\mathrm{d}h}. \tag{3}$$

In our conceptual model, we assume that the stress to hold the dyke open is provided by the deviatoric stress in the upper crust and the dyke is effectively a tank being filled by a conduit entering its base. Dividing $P$ into the pressure due to the weight of the crust above and below the base of the dyke, and considering the flow rate into the base of the dyke, $Q_D$, from (2)

$$\rho_u g D + \triangle\rho g L = \frac{8\nu L}{\pi r^4}Q_D \tag{4}$$

where $\rho_u$ is the average density of the upper crust (above the base of the dyke), $D$ is the depth to the base of the dyke and $\Delta\rho$ is the average difference between the densities of the lower crust and the magma.

Combining (3) and (4) gives

$$\frac{\mathrm{d}Q}{\mathrm{d}h} = \frac{-\rho_m Q_D}{\rho_u D + \triangle\rho L}. \tag{5}$$

Thus, if we can measure the inflow rate at the base of the dyke and the change of inflow rate with height in the dyke, we can determine the length of the conduit:

$$L = \frac{-\rho_m Q_D}{\triangle\rho\frac{\mathrm{d}Q}{\mathrm{d}h}} - \frac{\rho_u D}{\triangle\rho}. \tag{6}$$

Assuming values of $\rho_m = 2{,}700\ \mathrm{kg\ m}^{-3}$, $\rho_u = 2{,}700\ \mathrm{kg\ m}^{-3}$, $\Delta\rho = 300\ \mathrm{kg\ m}^{-3}$ and $D = 6$ km, and values estimated from our data of $Q_D = 32\ \mathrm{m}^3\mathrm{s}^{-1}$ and $\frac{\mathrm{d}Q}{\mathrm{d}h} = -0.0043\ \mathrm{m}^2\mathrm{s}^{-1}$ gives $L = 13$ km, which puts the source depth at an estimated 19 km. Note that this result is independent of the viscous pressure loss term and does not therefore depend on the conduit being cylindrical in shape (both $Q$ and $\mathrm{d}Q/\mathrm{d}h$ depend on the viscous term, which cancels when taking the ratio). Also note that the model does not require that $\rho_m$ is equal to $\rho_u$. Assuming $g = 9.8\ \mathrm{m\ s}^{-2}$ and $\nu = 100$ Pa s then gives an estimate for $r$ of 0.9 m. We thus infer that the cross-sectional area of the conduit is of the order of a few square metres. If the effective flow path is not comparable to that of a cylindrical conduit, then its cross-sectional area can be expected to be of similar magnitude, although viscous drag depends on the conduit shape[58].

## Data availability

A comprehensive collection of datasets utilized in this study is available at the Open Science Framework repository https://osf.io/n73cm/?view_only=97c944a29fe1471b8f663eec3d78fe54. This includes links to seven directories: GNSS Rinex files, wrapped interferograms, unwrapped interferograms, coherence files, geodetic modelling data, figure data and relocated earthquakes. The GNSS Rinex files directory contains the original Rinex files for all GNSS stations included in this study. The wrapped and unwrapped interferogram directories contain the interferograms displayed in Fig. 2 and Extended Data Fig. 2 in geotiff format, with metadata included as .xml files. The coherence files directory contains the corresponding coherence data related to the interferograms. The geodetic modelling data directory includes the input data for the geodetic inversion results displayed in Extended Data Figs. 5 and 6. The interferograms are provided in netcdf format and the GNSS data as a text file. The relocated earthquakes folder contains relative relocations of earthquakes following the procedure outlined within the Methods. Source data are provided with this paper for Figs. 2–4 and Extended Data Figs. 2, 3 and 8. Other figure data are found in the figure data folder at the Open Science Framework repository. We acknowledge use of data from the National Land Survey of Iceland for base maps for figures, available at https://www.lmi.is/is/landupplysingar/gagnagrunnar/nidurhal (licence: https://www.lmi.is/is/moya/page/licence-for-national-land-survey-of-iceland-free-data).

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

**Acknowledgements** This research was supported by the H2020 projects EUROVOLC and RISE funded by the European Commission (grant numbers 731070 and 821115, respectively), the Natural Environmental Research Council through the Centre for the Observation and Modelling of Earthquakes, Volcanoes and Tectonics (COMET), a IS-TREMOR Rannis grant (number 217738) and a MaDRe RanniS grant (number 228933-051). We thank IMO technicians and the Operations department, the IMO seismological monitoring team, the CAS as well as the ÍSOR for connecting eight of their seismic stations (currently running under the NASPMON project) to IMO's monitoring system to improve the detection and location of earthquakes and the University of Cambridge for access to data from seismic station odf. We thank the University of Iceland technicians for support and operation of the GNSS stations. We also thank the National Commissioner of the Icelandic Police and their Department of Civil Protection for organizing science-board meetings in connection with the unrest on Reykjanes Peninsula, and participants in these meetings for discussions and evaluations of the monitoring data. F.S. and H.G. acknowledge support from the University of Iceland Research Fund. We acknowledge reviews by D. Roman and F. Amelung that helped to significantly improve the paper. The GSNL Icelandic Volcanoes Supersite project supported by the Committee on Earth Observing Satellites is acknowledged for providing support to the InSAR monitoring/research in Iceland. The GMT open access software was used to produce a number of the figures in this paper.

**Author contributions** F.S. coordinated the writing of the paper and the development of the ideas presented. Geodetic modelling and interpretation was carried out by M.P. (uniform dyke opening models) and A.H. (distributed opening dyke model). Magma conduit flow

modelling was carried by A.H with input from F.S. GNSS observations and analysis were carried out by H.G., B.G.Ó. and S. Hreinsdóttir, and H.G. did the stress modelling. InSAR analysis and strain evaluation were carried by V.D. Compilation and calculation of $M_w \geq 4$ earthquakes and mapping of dyke segments with relatively relocated earthquakes was carried out by K.S.V. with contribution from S. Hjaltadóttir. Interpretation of seismicity and other seismic analyses were done by K.S.V., K.J., P.E., J.H. and T.Á. S.B. evaluated the implications of the study for monitoring at volcano observatories. All the authors contributed to evaluation of the modelling, discussion of the results and the writing of the paper.

**Competing interests** The authors declare no competing interests.

**Additional information**
**Correspondence and requests for materials** should be addressed to Freysteinn Sigmundsson.

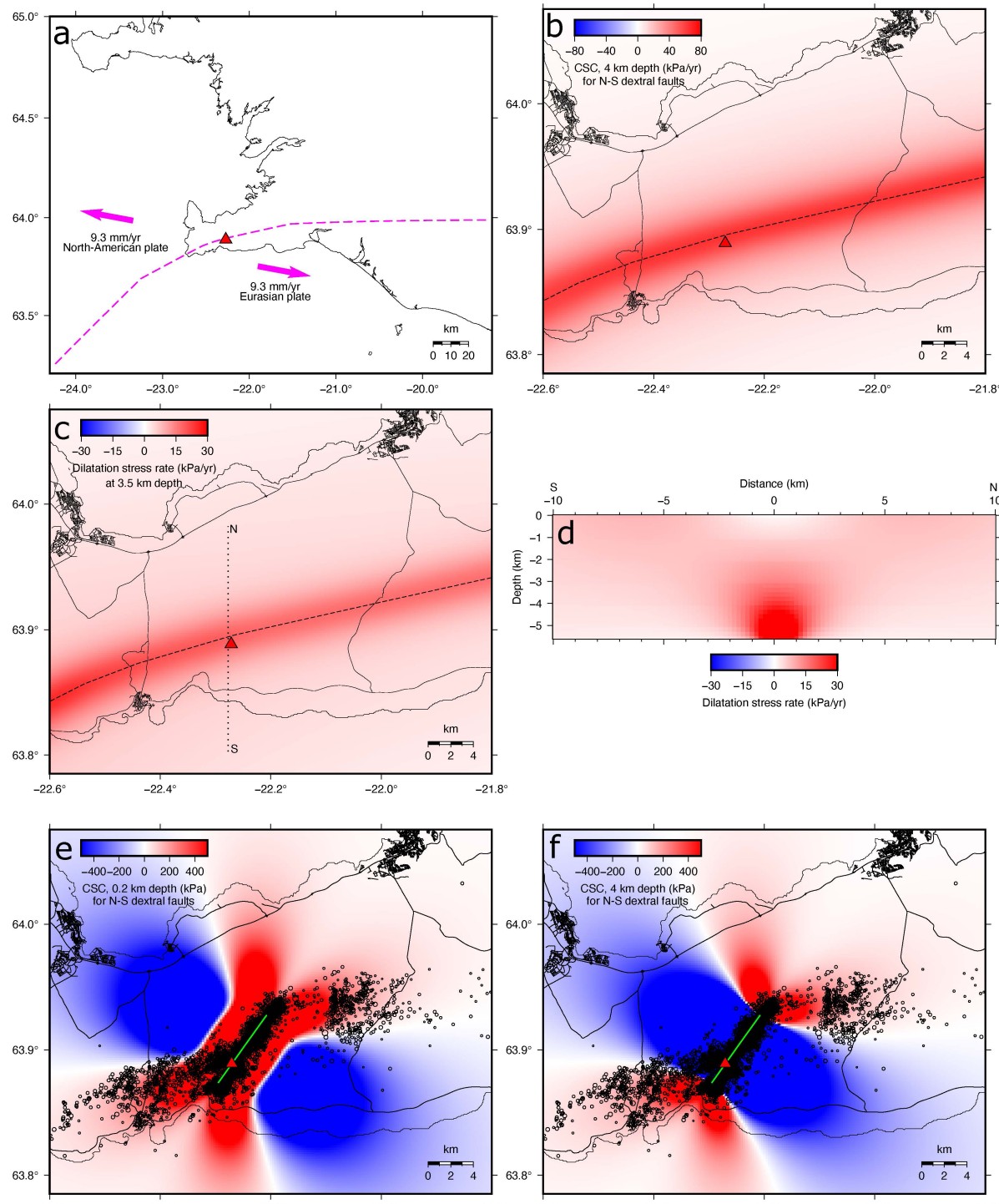

**Extended Data Fig. 1 | Plate bondary deformation model and associated stress. a**, Outline of plate boundary model segments used to calculate stress rates caused by the plate motion (see Methods; a locking depth of 6 km was assumed). Red triangle marks eruption site. **b**, Rate of Coulomb stress change (CSC) at 4 km depth for north-south striking, right-lateral, vertical strike-slip faults, due to plate motion. **c**, Rate of dilatational stress change due to plate

motion calculated at 3.5 km depth. **d**, North- south cross section of (c). **e**, Near-surface CSC for north-south, right-lateral, vertical strike-slip faults, caused by a simplified single-segment 2021 dike model. **f**, CSC at 4 km depth for north-south, right-lateral, vertical strike-slip faults, caused by same model as in (e). In (e) and (f), the earthquake locations plotted are the same as in Fig. 1a.

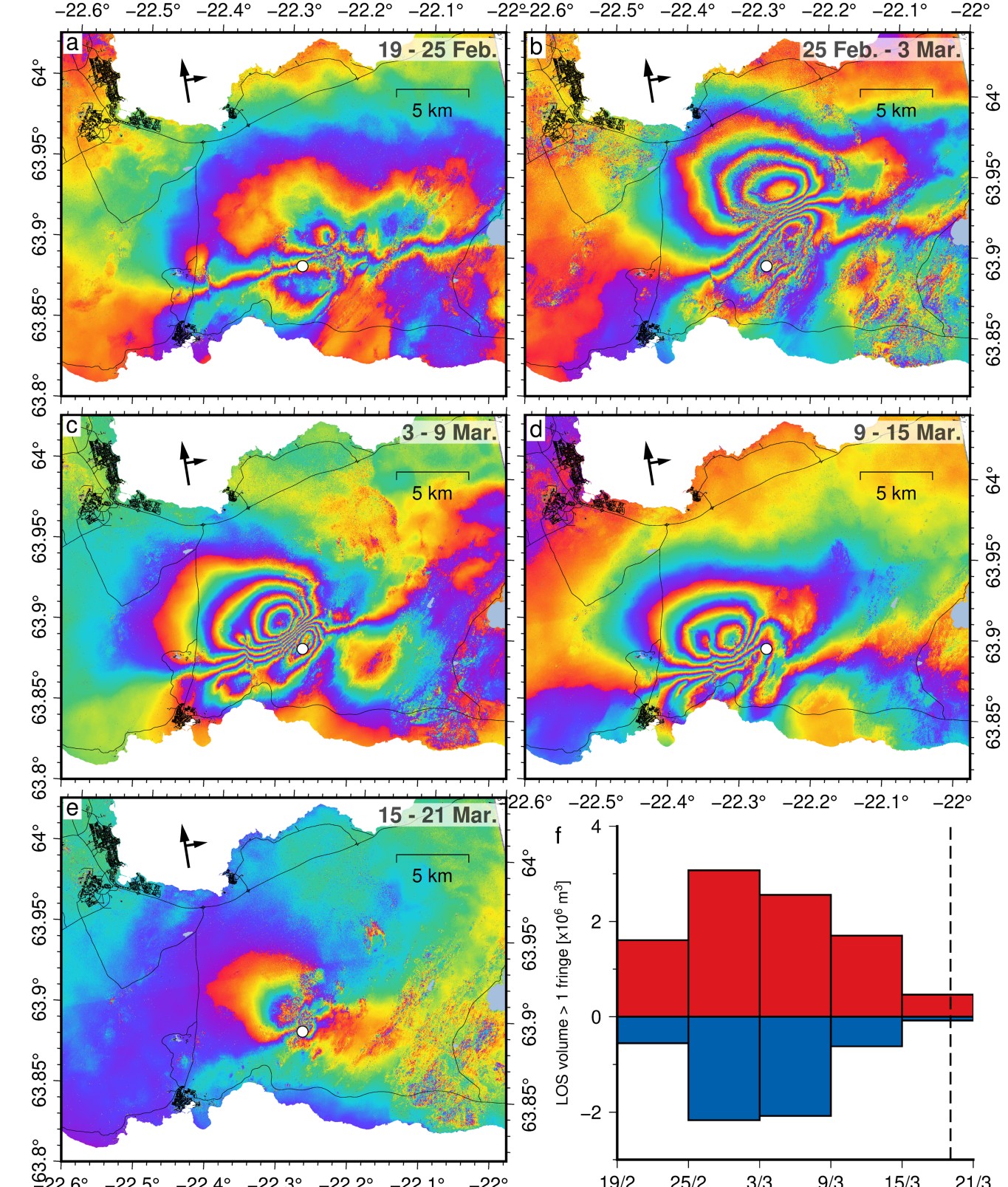

**Extended Data Fig. 2 | Spatial evolution of deformation.** Same as Fig. 2, for ascending satellite track T16 (acquired 18:59 local time).

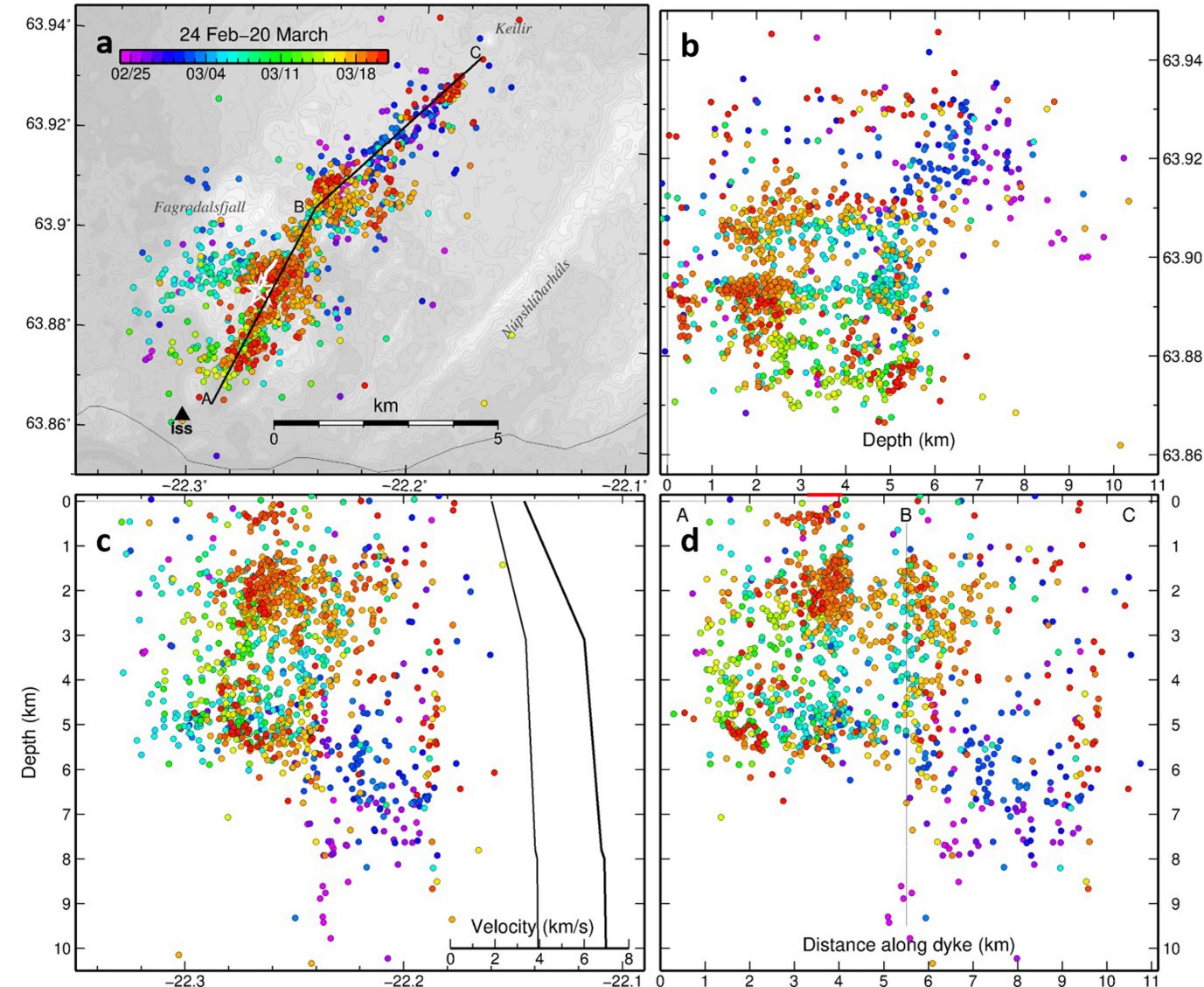

**Extended Data Fig. 3 | Relocated Earthquakes. a**, Relatively relocated earthquakes 24 February to 20 March along the dyke (magnitude range 0.5 ≤ M < 1.4), shown in map view and vertical-sections. **a**, Epicentres (circles) coloured according to origin time. White star on the map shows location of the first eruptive fissure and the range of eruptive vents (white line) is shown. The nearest seismic station (black triangle) is also shown. **b**, Focal depths vs. latitude, viewed from the east. **c**, Focal depths viewed from the south. Velocity model used for relocations is also shown (thin, black line for S-wave velocity, thick for P-wave velocity). **d**, focal depths along the two dyke sections AB and BC (shown on the map). Red line along upper x-axis shows the extent of the eruption vents. The vertical cross sections show that the seismicity in the last three days prior to the eruption (red coloured dots) is largely concentrated in two clusters, at depths of 1-2.5 km and ~0.5 km under the eruption site suggesting that magma had already started to accumulte near the surface.

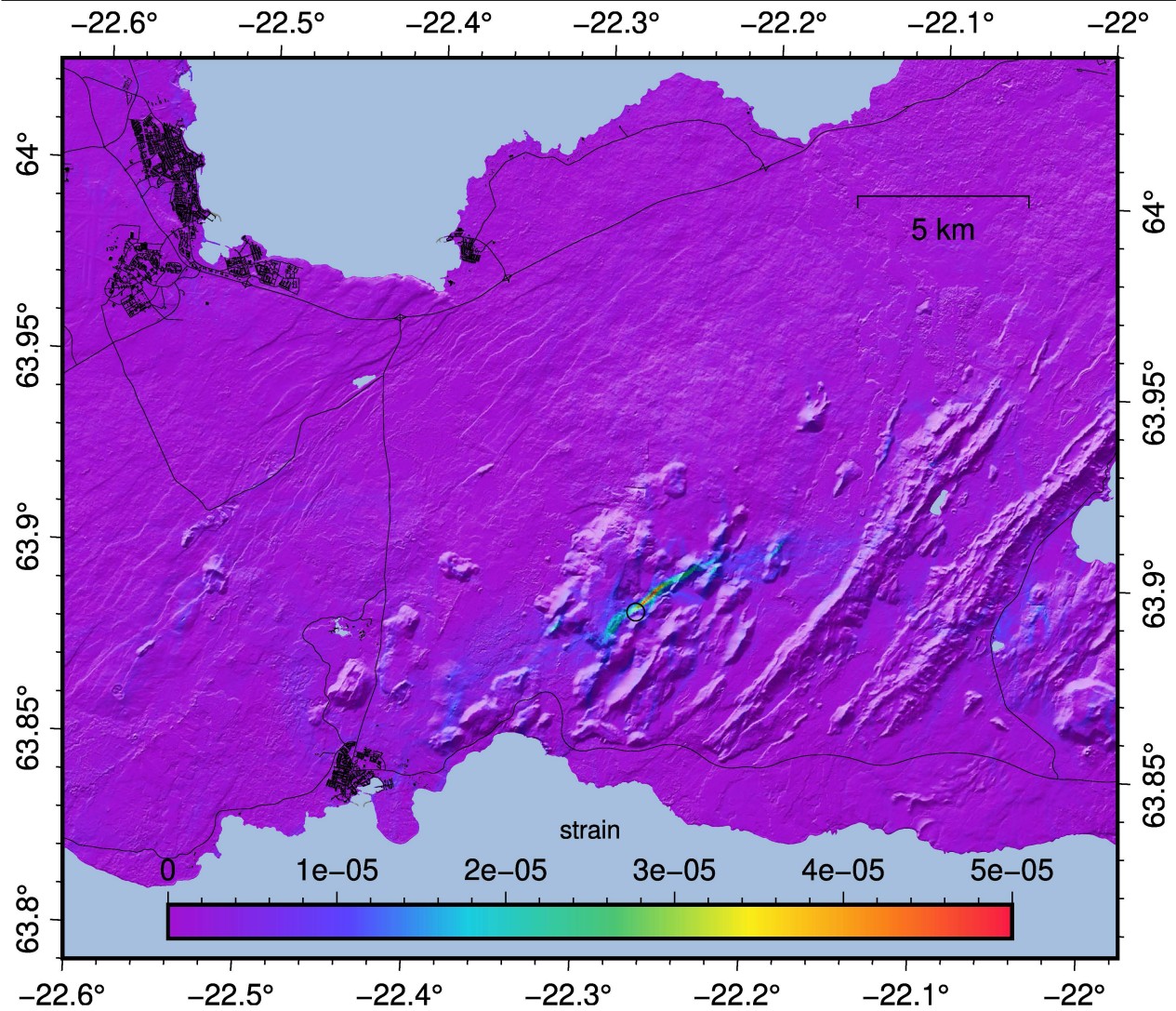

**Extended Data Fig. 4 | Yearly strain 2015-2018.** Second invariant of yearly strain[64] in 2015-2018 inferred from InSAR analysis of ascending and descending Sentinel tracks[41], ignoring contribution of northward component of horizontal displacement (< 5 mm/yr). A median filter (11-pixel wide) was applied to the grid before calculating the strain. High strain rate lines up the central axis of the plate boundary. The 2021 eruption (circle) occurred within this area.

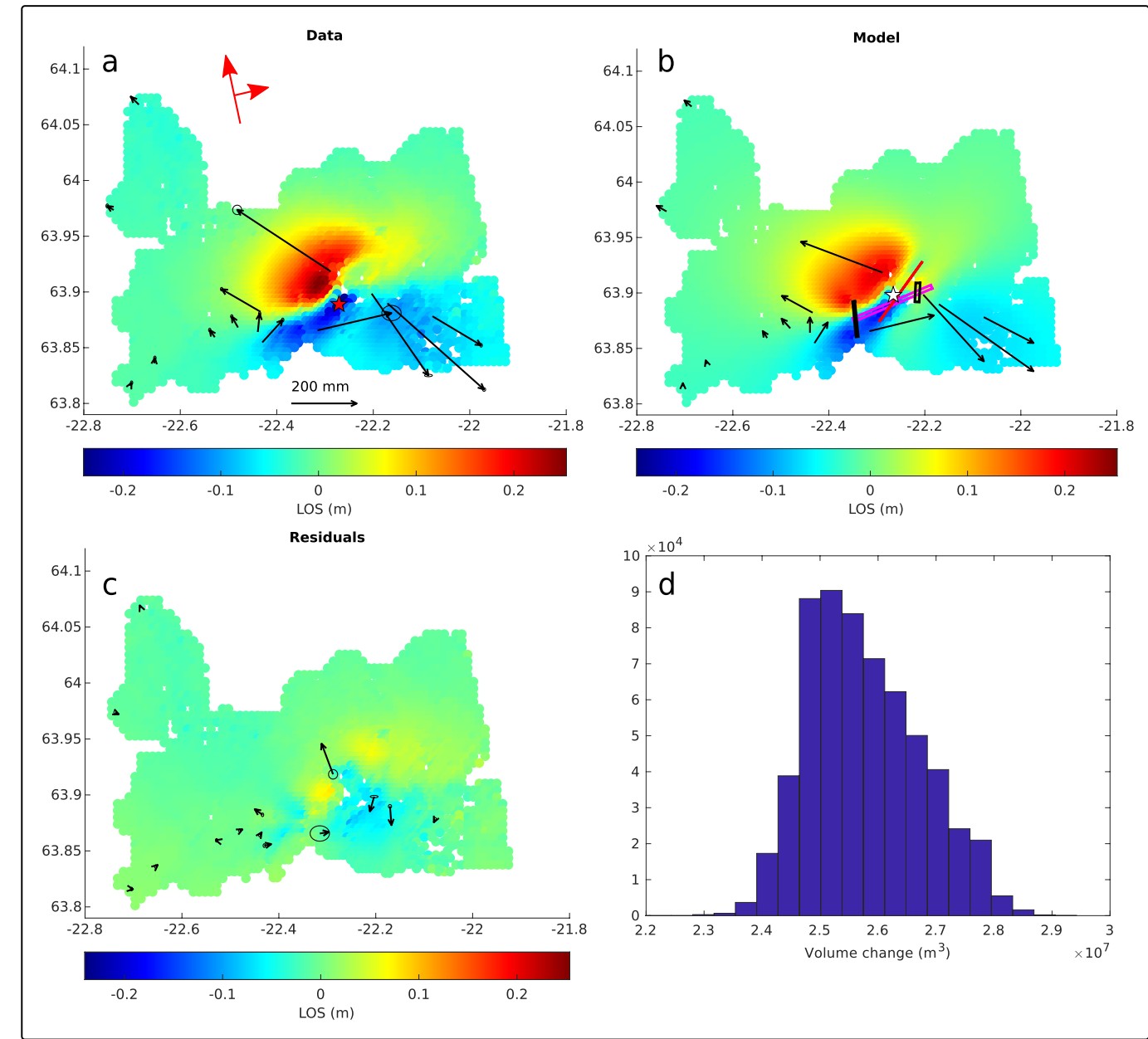

**Extended Data Fig. 5 | One-segment dyke model. a**, Input data to inversion (GNSS observations and ascending T16 interferogram covering the period 19 February to 21 March. **b**, Modelled horizontal GNSS displacements and modelled LOS displacements for T16 interferogram. **c**, Residual GNSS and LOS displacements. **d**, Inferred probability density function of volume change (magma inflow). In panel a, the red star shows the location of the eruption site and the red arrows indicate the satellite heading and look direction. In panel b, the red line shows the location of the dyke, the magenta line is the dislocation source along the plate boundary. The near-vertical black lines represent the modelled faults and the white star displays the location of the deflation source.

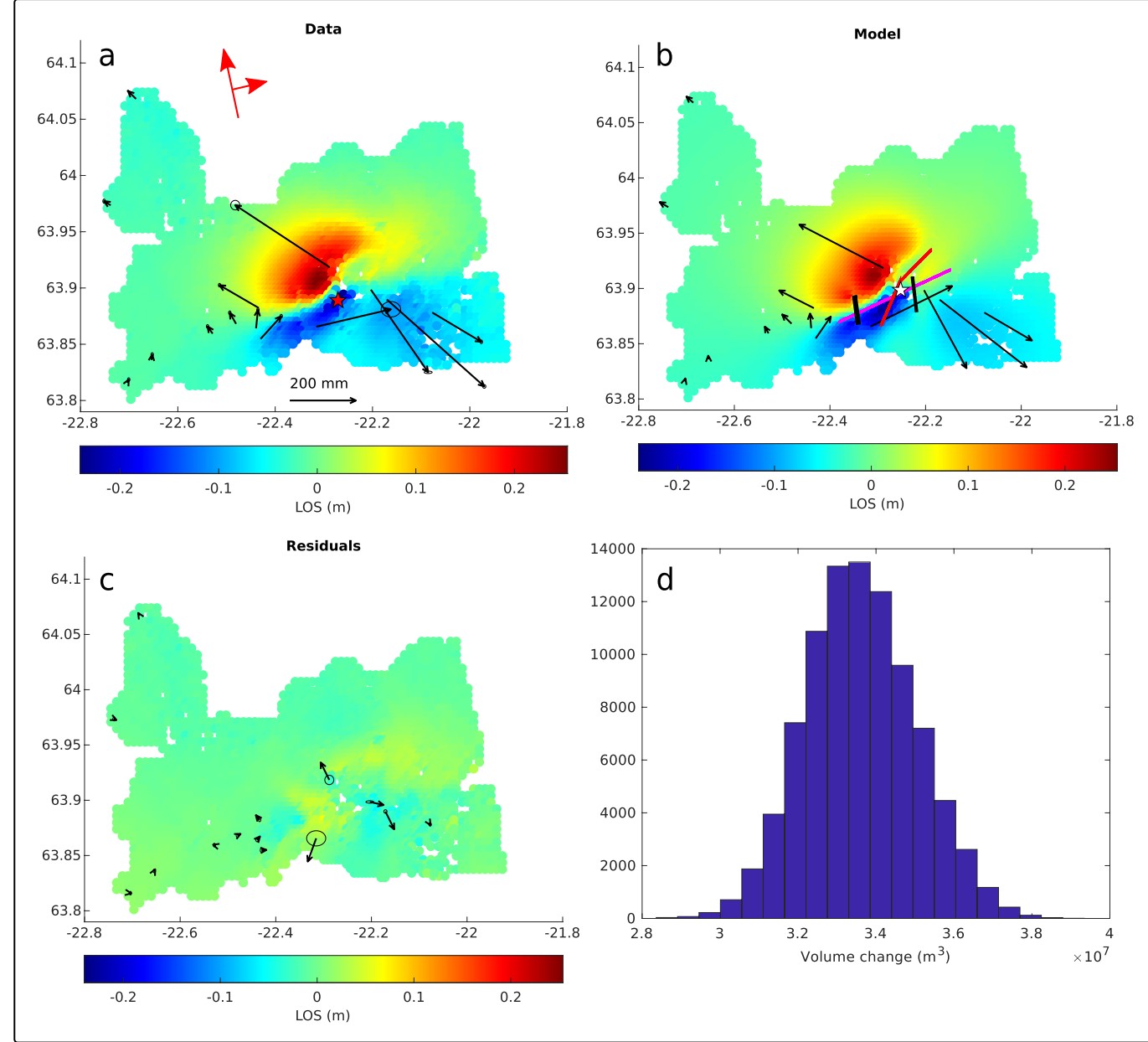

**Extended Data Fig. 6 | Two-segment dyke model with multiple patches.**
**a**, Input data to inversion (GNSS observations and ascending T16 interferogram covering the period 19th February to 21st March. **b**, Modelled horizontal GNSS displacements and modelled LOS displacements for T16 interferogram. **c**, Residual GNSS and LOS displacements. **d**, Inferred probability density function of volume change (magma inflow). In panel a, the red star shows the location of the eruption site and the red arrows indicate the satellite heading and look direction. In panel b, the red line shows the location of the two-segment dyke, the magenta line is the dislocation source along the plate boundary. The near-vertical black lines represent the modelled faults and the white star displays the location of the deflation source.

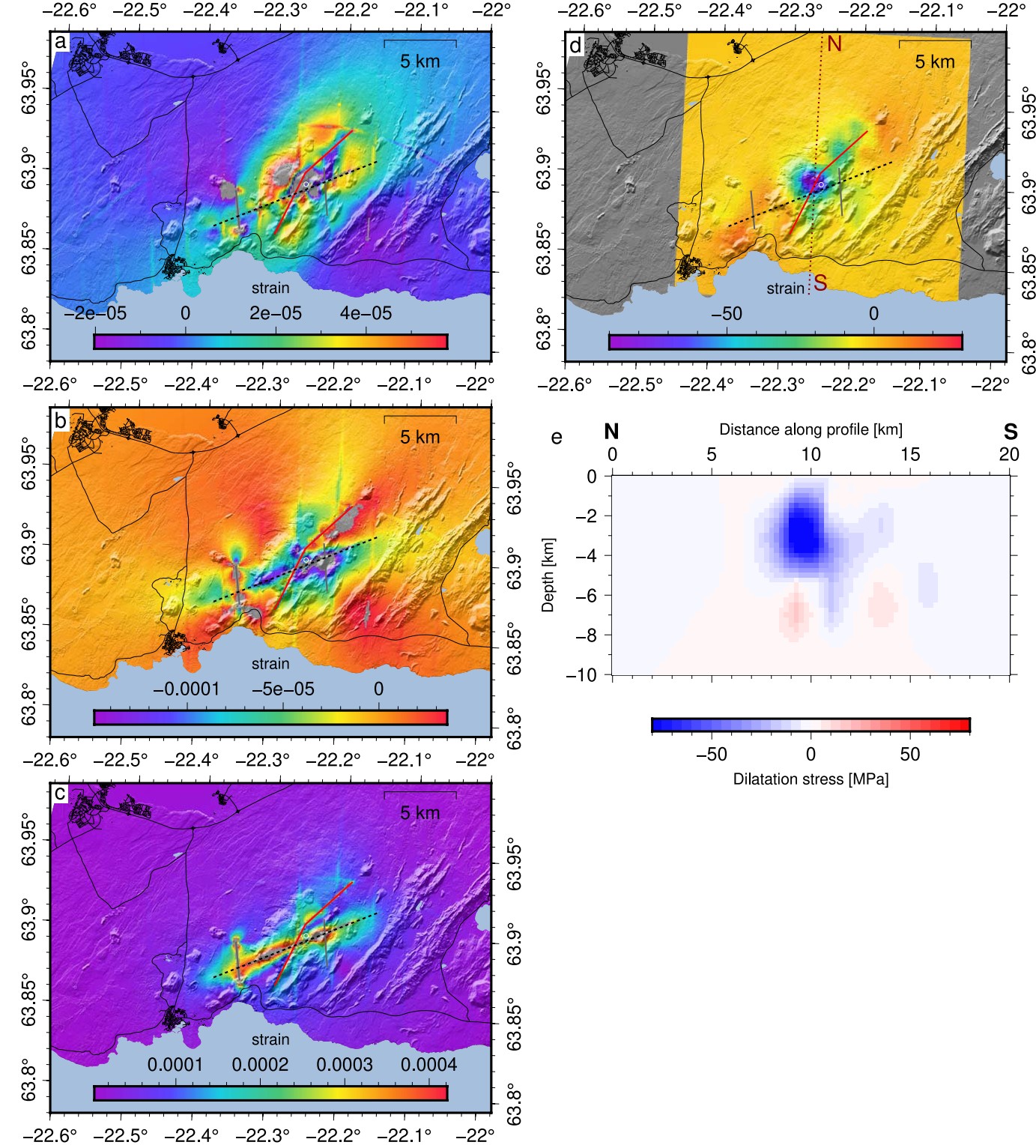

**Extended Data Fig. 7 | Stress and strain.** Surface strain according to modelled deformation sources (see Fig. 4), calculated in the same manner as Drouin et al.[63]. **a**, areal strain. **b**, shear strain and **c**, second invariant of strain.

Dilatational stress change at 3 km depth in map view (**d**), and on a vertical cross section (**e**), at location shown in **d**. The dilatational stress is calculated in the same manner as for the plate motion model shown in Extended Data Fig. 1.

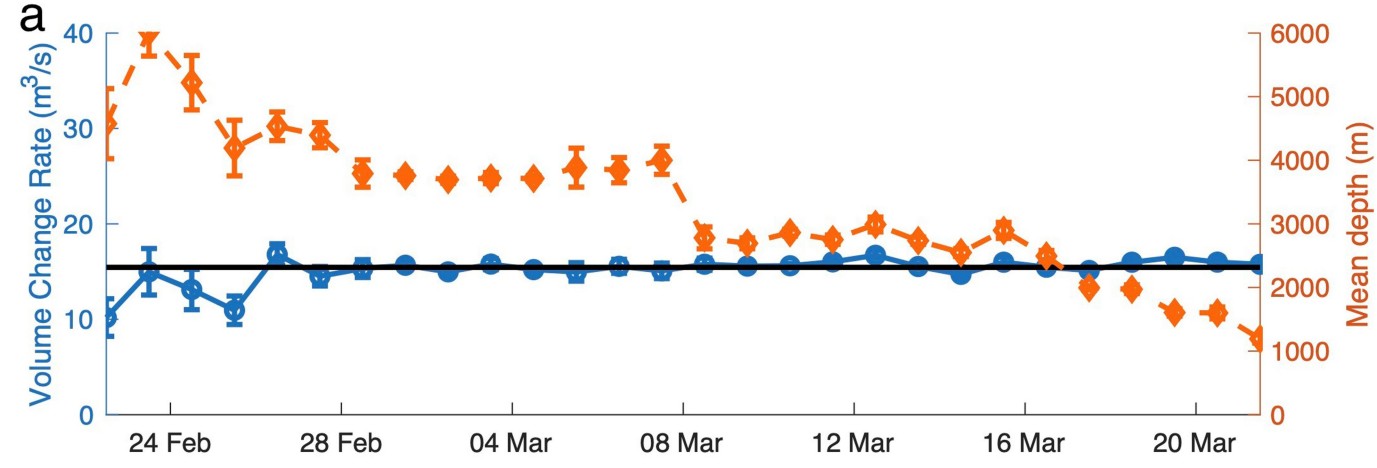

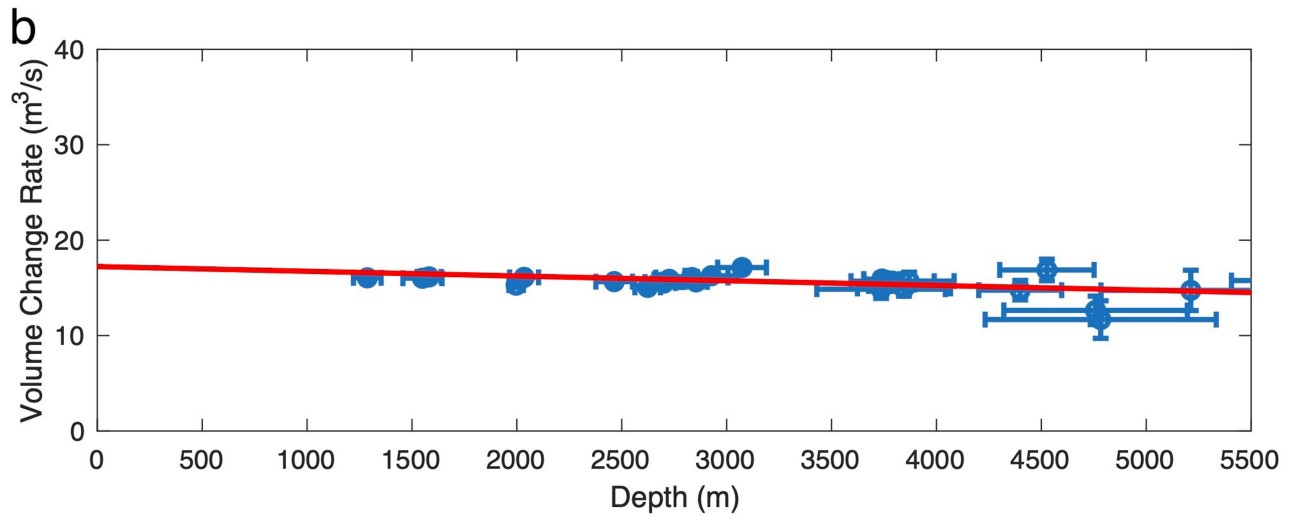

**Extended Data Fig. 8 | Results of an inversion simulation test. a**, Daily volume change rate and mean depth of the volume change for data simulated with constant volume change and shallowing depth. The black line indicates the true volume change rate and error bars represent 1 standard deviation.

**b**, Values from **a** plotted against each other. The red line is the best fitting line using least squares. The correct volume change rate is retrieved despite varying depth of magma emplacement.

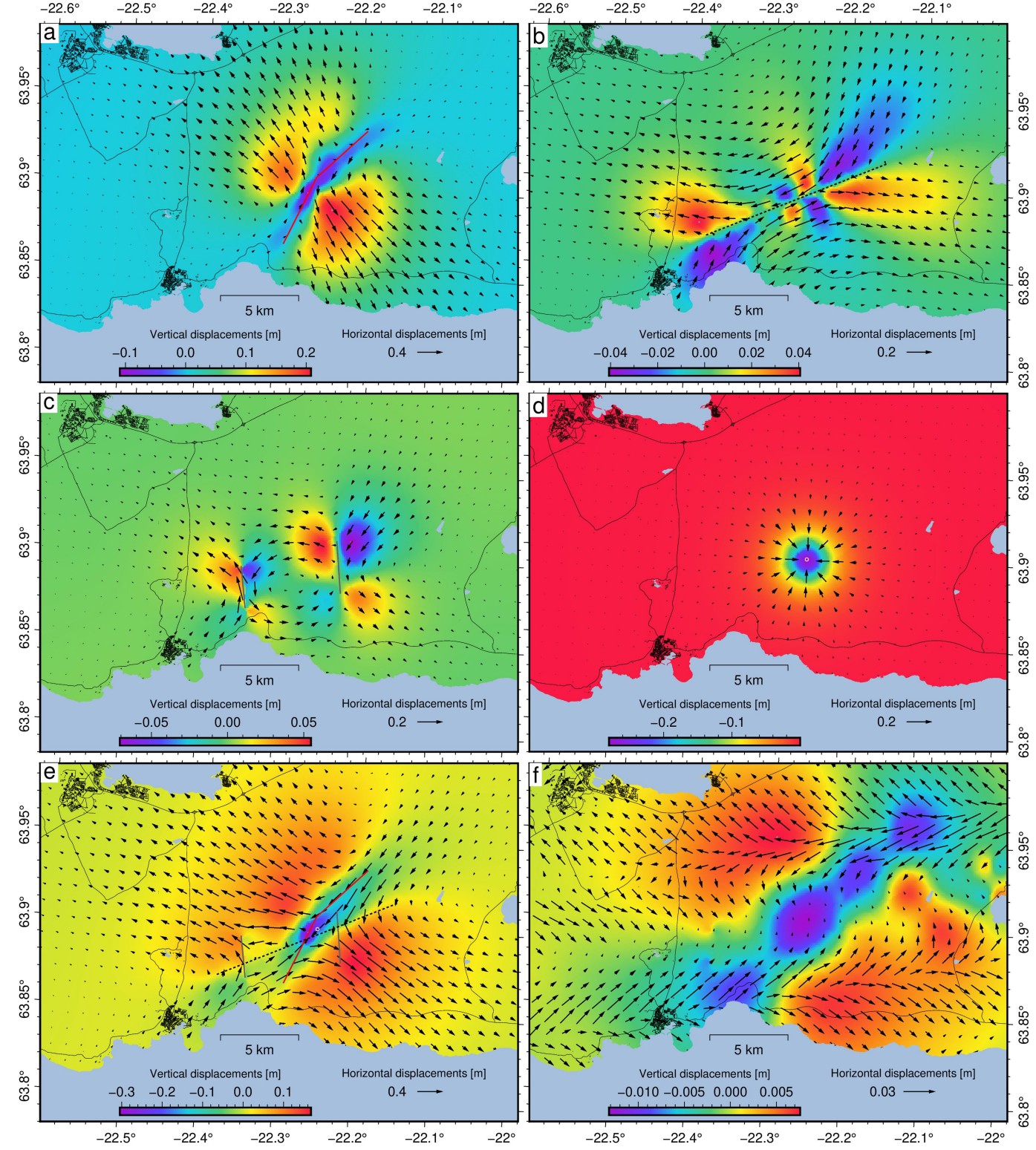

**Extended Data Fig. 9 | Modelled deformation.** Panels show deformation due to individual sources in final deformation model: **a**, dyke; red line. **b**, plate boundary; dashed line. **c**, two largest earthquake faults; grey lines. **d**, deflation source; white circle. **e**, all sources combined; same as Fig 4a. The last panel (**f**) shows modelled cumulative surface deformation induced by earthquakes M > 4, except the two largest ones (Methods).