## [Peer Review File · Nature]

Manuscript Title: Deformation and seismicity decline before the 2021 Fagradalsfjall eruption, Iceland

Reviewer Comments & Author Rebuttals

Reviewer Reports on the Initial Version:

Referee #1:

Review of the paper “Deformation and seismicity decline preceding a rift zone eruption” by Sigmundsson et al.

This paper presents the intriguing observation of a rifting event in Iceland that leads to an eruption only after a significant amount of tectonic stress is released (4 weeks after the beginning of the unrest period). This is an important observation that is worth publishing in Nature. The implication is that there are volcanic eruptions not preceded by increases in seismicity and deformation, i.e. increases in seismicity and deformation is not a necessary precursor for an eruption. The frequently used failure forecast method (FFM) would not work. This has important implications for volcanic hazards, as it suggest that an eruption may occur weeks or months after a rifting episode has subsided.

The results of this study are highly relevant for the current unrest period at Mauna Loa volcano in Hawaii. It is thought that the next eruption will be heralded by significant seismicity and deformation. But maybe not?

The paper needs several additions/clarifications before it can be published.

1. The paper contains strain calculation, but wouldn't it make sense to present stress calculations? What is the state of stress at the location of the eruption after the earthquake and dike injection? The stress state above the dike must be tensional. So the injection of an upward dike is not a big surprise. How much tension was imparted above the dike? This eruption appears to have similarities with the so-called 'passive' dike intrusions/eruption at Kilauea (in response to tensional stress and not to forceful magma intrusion), something what could be discussed.

2. if I understood correctly, the magma was sitting for a while in the dike before it propagated towards the surface to erupt. This appears to be an opportunity to constrain the pressure conditions required for upward dike propagation. Is the sequence consistent with delayed, decompression-induced exsolution of volatiles from the magma that eventually created a high enough pressure to break the rock?

3. I did not get convinced about the correlation between magma flow rate and the depth of magma emplacement represented in Fig. 3c. It is well known that geodetic inversions have trade-offs between the depth and opening of an elastic dislocation. The authors need to demonstrate that the inferred linear relationship does not represent this trade-off.

Presentation and clarity.

- The paper would benefit from a sketch depicting the sequence of events. I had a hard time to understand when which event/process occurred.
- Several Figs lack a),b),c),d) etc
- Is this manuscript considered as a letter or as an article? It can be easily shortened to letter format.

Minor comments:

- The title contains "rift zone eruption". Please define. Do you mean an eruption similar to a spreading center eruptions, most of which occur submarine? Do you consider the Reykjanes Peninsula as a spreading center?
- Abstract: date of Mw5.64 quake unclear
- Please comment some more on the M5.64 quake. Was this a regular quake or triggered by fluid (magma) intrusion into the fault plane?
- Is it necessary to talk in so much detail on the failure forecast method? I expected the equation to be used quantitatively, but it is not.
- That the Reykjanes Peninsula rift is oblique is not clear from the Figs. Some plate motion vectors might clarify this.
- Is the oblique spreading important for the precursory deformation decline? The emphasis on obliqueness at the beginning of the paper suggests this, but the paper does not return to any obliqueness.
- Please motivate the strain analysis: What is better resolved by looking at strain compared to displacements? As strain is the derivative it contains less information than displacement.
- Line 55: highly oblique : Give the angle?
- line 64: Strain accumulation as predicted by a rifting, shear or by combined model? Stating the locking depth suggest strain accumulation along NS-oriented strike-slip faults?
- line 126: earthquake deformation by the Feb 24 earthquake only?
- line 131: the strike of the plate boundary was not stated.
- Fig. 2 lower right is not explained. What is LOS volume?
- Fig.3a: blue cross/star not explained
- line 135: not clear where and during which periods this localized subsidence occurs
- line 143: add reference to Methods
- line 148: define 'mean depth of magma emplacement'. Is this the center of an opening dislocation?
- line 150: could the relationship between magma flow rate and depth be an inversion artifact representing a trade-off between these parameters?
- Fig. 4a: the surface displacements towards the source are striking (in the green area). Some explanation of how the superposition of various sources create such a pattern would help (e.g. a figure in the supplement showing the displacements of the individual sources that are summed up).
- line 160: explain why you take a derivative in cartesian directions and not in parallel and perpendicular directions to the plate boundary.

An aside:

- Line 197: Please be careful with the statement on the role of magma buoyancy. It is true that is it important when the rock is already broken but whether it can break the rock is contested and does

not add much to the problem in question. Urbani et al (2018) and Gregg et al. (2015) come to the conclusion that it plays a subordinate role for dike injection and propagation. To what I recall, in your Sigmundsson et al [2020 Nat.Comm] paper to which you are referring here, you are implicitly assuming lithostatic pore fluid pressure when you say that failure occurs once the tensile strength is reached (Albino et al., 2018), which is a questionable assumption (see Grosfils-Gudmundsson controversy).

Falk Amelung

Gregg, Patricia M., Eric B. Grosfils, and Shanaka L. de Silva. "Catastrophic caldera-forming eruptions II: The subordinate role of magma buoyancy as an eruption trigger." *Journal of Volcanology and Geothermal Research* 305 (2015): 100-113

Urbani, S., V. Acocella, and E. Rivalta. "What drives the lateral versus vertical propagation of dikes? Insights from analogue models." *Journal of Geophysical Research: Solid Earth* 123, no. 5 (2018): 3680-3697.

Referee #3:

This is a novel and important study of patterns of unrest preceding the 2021 eruption at Fagradalsfjall, Iceland. The study makes a convincing case for a period of decreasing deformation following high rates of deformation and immediately preceding the eruption (see my comments on seismicity rates below), and for a model of tectonic strain release to explain the observed temporal patterns of unrest rates. The study provides strong evidence in support of an emerging paradigm (in contrast to the FFM paradigm) of short-term precursory quiescence before eruptions, and thus ultimately warrants publication in *Nature*. However, I have several concerns regarding presentation and interpretation that should be addressed before the paper is acceptable for publication.

Major concerns:

1. The motivation for the study and its importance as given in the Introductory section is relatively weak, and could be greatly strengthened by enumerating additional prior examples of short-term precursory quiescence worldwide in a larger range of contexts (stratovolcanic eruptions (e.g., Pinatubo, Redoubt), phreatomagmatic eruptions (e.g., Suwanose-jima, Telica). As it stands only one other example, from Eyjafjall, Iceland, is mentioned in the introduction.

2. My major concern is that the purported decline in seismicity rate (as shown in Figure 1b, top panel) is not entirely convincing. There is a clear decline in the rate of large-magnitude earthquakes (and thus cumulative magnitude) as shown in the middle panel. However, the slope of the cumulative number of earthquakes curve is minimal at best, indicating the possibility that the large magnitude earthquakes and their aftershocks dominate the trend. This is a challenging issue to assess. Perhaps one means of addressing it would be to examine the rate of lower magnitude earthquakes through time (what is the magnitude of completeness for the catalog?). Or to attempt to deconvolve mainshock-aftershock sequences from the catalog by, e.g., removing the periods immediately following mainshocks from the plot.

3. (Lines 154-155 and Methods) - I do not think that a model of a cylindrical conduit is appropriate, as the magma transport structure is clearly a fissure all along its depth.

4. (Statement beginning on L199). I strongly disagree with the characterization of the initial Fagradalsfjall seismicity as 'distal volcano-tectonic'. Distal volcano-tectonic earthquakes are earthquakes occurring in a volume of rock that is separated in distance from the intruding magma, which is not the case for the Fagradalsfjall earthquakes which all occur proximal to an intruding dike. The fact that the ultimate vent for the Fagradalsfjall eruption is some distance from the early (proximal to intruding dike) earthquakes does not meet the (arguably ambiguous) definition of 'distal VTs' given in ref 25 (from its source paper - White and McCausland 2016). Given the numerous issues with the published definition of the term 'distal VT' and their characterization and interpretation ((see discussion in Wauthier et al. 2016, EPSL; and Meyer et al. 2021, JVGR, for example), I recommend that this term and references to the White and McCausland studies be avoided entirely.

5. I agree with the general interpretation (paraphrasing L208-211) that progressive lowering of crustal stress in rifting environments will exert a first-order control on temporal patterns of precursory seismic moment release and inflation. It would be useful to discuss this interpretation in the context of the large body of work on active vs passive rifting in East Africa (e.g., Ebinger et al. 2010; Keir et al. 2013; Pallister et al. 2010, and many others). For example, does this interpretation provide insight into long-term intrusive/eruptive ratios in rift settings, for example?

Minor comments/edits:

L80 - "eastward" -> "northeastward"?

L89 - "N295E" - Legend for Figure 1b (lower panel) says "325 degrees"?

L92-94 - I think the sentence beginning with "When" can be omitted.

L94 - "It is also evident.." -> "It is also evident (from InSAR)..."

L115-117 - It seems that the seismicity to W is early, and occurs before the south segment seismicity.

L158-159 - The sentence as written is unclear - does this mean you're assuming that all of the sources (including dike) were active for a long time prior to eruption, or is stress accumulated over a long time? To clarify I suggest rewording as follows: "the strain and stress fields due to all the sources combine in a constructive manner to release stress accumulated by plate movements over a long time period prior to the eruption."

L413 - "47,000"

L414 - "7,500"

L416 - "Czech"

L661 - "dominating fault structure" -> "dominant fault orientation and sense of slip"

Fig 1a - The color bar dates are confusing (red is apparently ~2 months after eruption onset). Also meaning of "50 mm" is confusing.

Fig 2 - Which pixels are used for calculations in F? (Same issue applies to EF 1)

Fig 3 - What is the meaning of the blue asterisk?

Fig 3b - I suggest reversing the blue axis (so that 0 is on the bottom and 40 is on the top), as the

current presentation is non-intuitive.

Thank you and I look forward to reading the final version of this important manuscript.

Kind regards,
Diana Roman

Author Rebuttals to Initial Comments:

Point-by-point response to the referees

We thank the reviewers for all their important and highly relevant comments. See below responses (normal font) to all the comments raised (*italics font*). Line numbering referred to below corresponds to line numbers in the clean version (no track changes) of the revised manuscript.

Referee #1:

Review of the paper “Deformation and seismicity decline preceding a rift zone eruption” by Sigmundsson et al.

This paper presents the intriguing observation of a rifting event in Iceland that leads to an eruption only after a significant amount of tectonic stress is released (4 weeks after the beginning of the unrest period). This is an important observation that is worth publishing in Nature. The implication is that there are volcanic eruptions not preceded by increases in seismicity and deformation, i.e. increases in seismicity and deformation is not a necessary precursor for an eruption. The frequently used failure forecast method (FFM) would not work. This has important implications for volcanic hazards, as it suggest that an eruption may occur weeks or months after a rifting episode has subsided.

We thank the reviewer for his important comments that have helped to improve the manuscript.

The results of this study are highly relevant for the current unrest period at Mauna Loa volcano in Hawaii. It is thought that the next eruption will be heralded by significant seismicity and deformation. But maybe not?

We now discuss unrest at Mauna Loa in the final paragraph of the manuscript.

The paper needs several additions/clarifications before it can be published.

1. The paper contains strain calculation, but wouldn't it make sense to present stress calculations? What is the state of stress at the location of the eruption after the earthquake and dike injection? The stress state above the dike must be tensional. So the injection of an upward dike is not a big surprise. How much tension was imparted above the dike? This eruption appears to have similarities with the so-called 'passive' dike intrusions/eruption at Kilauea (in response to tensional stress and not to forceful magma intrusion), something what could be discussed.

Stresses in the crust are indeed important and we have added more evaluation and discussion of these. We added a model of stresses generated by the oblique plate spreading processes taking place in the area prior to the events, so these can be put in context with the stress change due to the 2021 events. See Extended Data Figs. 1 and 7, and discussion in lines 173-180.

A comparison to passive dike intrusions/eruptions at Kilauea is highly relevant, we have added this at the beginning of the Implications section.

2. if I understood correctly, the magma was sitting for a while in the dike before it propagated towards the surface to erupt. This appears to be an opportunity to constrain the pressure conditions required for upward dike propagation. Is the sequence consistent with delayed, decompression-induced exsolution of volatiles from the magma that eventually created a high enough pressure to break the rock?

Without knowing the pre-existing stress field, it is not possible to unambiguously determine the overpressure in the dyke, making it difficult to answer the question on pressure conditions required for propagation. And with continuing inflow of magma, it is also difficult to disentangle any pressure change due to volatile exsolution. With more sophisticated modelling and additional constraints (e.g. from studies of volatiles and degassing processes), perhaps it would be possible to address these questions, but we see this as beyond the scope of this paper.

3. I did not get convinced about the correlation between magma flow rate and the depth of magma emplacement represented in Fig. 3c. It is well known that geodetic inversions have trade-offs between the depth and opening of an elastic dislocation. The authors need to demonstrate that the inferred linear relationship does not represent this trade-off.

We paid special attention to this valuable comment. We now validate our results with a simulation test as explained in Methods (lines 543-552) and new Extended Data Figure 8.

Presentation and clarity.

- The paper would benefit from a sketch depicting the sequence of events. I had a hard time to understand when which event/process occurred.

We have improved presentation and clarity throughout the paper. In particular, we have paid attention to improving description of sequence of events throughout the text and in revised Figure 1. Relocated earthquakes also show the time progression. We have not added a separate sketch depicting the sequence of events, since we did not come up with a good idea how that could provide more information than already now given in the manuscript.

- Several Figs lack a),b),c),d) etc

We have added labels to all panels.

- Is this manuscript considered as a letter or as an article? It can be easily shortened to letter format.

We here followed the editor's guideline for formatting, and have ensured that our revisions did not lengthen the main text of the manuscript by more than ~200 words.

Minor comments:

- The title contains "rift zone eruption". Please define. Do you mean an eruption similar to a spreading center eruptions, most of which occur submarine? Do you consider the Reykjanes Peninsula as a spreading center?

Wording in first paragraph of main text has been adjusted to make it clear that the Reykjanes Peninsula has been, in Icelandic nomenclature of geoscientists, classified as an oblique rift zone, and that volcanic systems as defined in Iceland are comparable to submarine spreading centers.

- Abstract: date of Mw5.64 quake unclear

Because of word limit – we decided not to add this information to the abstract. The reader will find the date of this earthquake in the second paragraph of the chapter on “Pre-eruptive seismicity and deformation”.

- Please comment some more on the M5.64 quake. Was this a regular quake or triggered by fluid (magma) intrusion into the fault plane?

We extended the coverage of the M5.64 quake and events before and following (see lines 81-91). There is some uncertainty in exact nature of the event. However as it was preceded by highly repeating self-similar microearthquakes, often observed in the context of fluid migration. We find it likely that the earthquake was indeed triggered by magma movements.

- Is it necessary to talk in so much detail on the failure forecast method? I expected the equation to be used quantitatively, but it is not.

The coverage of the failure forecast method has been shortened and equations removed. This also helps to shorten the text.

- That the Reykjanes Peninsula rift is oblique is not clear from the Figs. Some plate motion vectors might clarify this.

Plate motion vectors have been added to Fig. 1 and Extended Data Fig. 1, and description of the obliqueness is now found in lines 58-59.

- Is the oblique spreading important for the precursory deformation decline? The emphasis on obliqueness at the beginning of the paper suggests this, but the paper does not return to any obliqueness.

We have not emphasized the obliqueness in the general conclusions, e.g in the implications when we say ... Our observations suggest release of tectonic stress followed by decline in deformation and seismicity rate may be a characteristic precursory activity anticipated for a certain class of eruptions. We try to emphasize that the release of tectonic stress is important. How this exactly happens can vary from one rift to another. In our case, however, the obliqueness (that we present better) is a fundamental factor determining the style of activity (the complicated sources of deformation).

- Please motivate the strain analysis: What is better resolved by looking at strain compared to displacements? As strain is the derivative it contains less information than displacement.

It is important to understand both displacements and strain, and we show both. Strain highlights the areas of largest gradients in displacement fields, and thus relates to stress that depends on the strain. The shearing mentioned in lines 183-184 is an important part of the stress release process during the events, and this we have kept unchanged.

- Line 55: *highly oblique* : Give the angle?

We now describe the obliqueness.

-line 64: *Strain accumulation as predicted by a rifting, shear or by combined model? Stating the locking depth suggest strain accumulation along NS-oriented strike-slip faults?*

This has been clarified in new Extended Data Fig. 1

-line 126: *earthquake deformation by the Feb 24 earthquake only?*

We have reworded this as co-seismic deformation and it will be clear for readers this applies to more than the Feb 24 earthquake.

- line 131: *the strike of the plate boundary was not stated.*

The strike of the central axis of the plate boundary is now given in the beginning of the chapter on “Pre-eruptive seismicity and deformation”.

- Fig. 2 lower right is not explained. What is LOS volume?

We have improved description and hope it is understandable now for a general reader.

- Fig.3a: *blue cross/star not explained*

Has been added.

-line 135: *not clear where and during which periods this localized subsidence occurs*

We now explain it is a gradually contracting point source of pressure throughout the events

-line 143: *add reference to Methods*

Reference to methods has been added to Fig. 3 caption.

- line 148: *define ‘mean depth of magma emplacement’. Is this the center of an opening dislocation?*

Now done in Methods.

- line 150: *could the relationship between magma flow rate and depth be an inversion artifact representing a trade-off between these parameters?*

See response to main comment 3 above. Our conclusion is that it is not an artifact.

- Fig. 4a: *the surface displacements towards the source are striking (in the green area). Some explanation of how the superposition of various sources create such a pattern would help*

(e.g. a figure in the supplement showing the displacements of the individual sources that are summed up).

We have added this in the present Extended Data Figure 9.

- line 160: explain why you take a derivative in cartesian directions and not in parallel and perpendicular directions to the pate boundary.

Explanation added. It is because it is possible to compare it directly to InSAR observations now explained in line 181.

An aside:

- Line 197: Please be careful with the statement on the role of magma buoyancy. It is true that is it important when the rock is already broken but whether it can break the rock is contested and does not add much to the problem in question. Urbani et al (2018) and Gregg et al. (2015) come to the conclusion that it plays a subordinate role for dike injection and propagation. To what I recall, in your Sigmundsson et al [2020 Nat.Comm] paper to which you are referring here, you are implicitly assuming lithostatic pore fluid pressure when you say that failure occurs once the tensile strength is reached (Albino et al., 2018), which is a questionable assumption (see Grosfils-Gudmundsson controversy).

We have evaluated this note carefully. Our conclusion is that the statement on magma buoyancy is careful enough. We agree that there is controversy regarding the role of magma buoyancy in breaking rock. This issue is not dealt with in our paper. The statement in original manuscript line 197, that we keep unchanged, relates to effects of buoyancy in driving magma up already established channel.

Falk Amelung

Gregg, Patricia M., Eric B. Grosfils, and Shanaka L. de Silva. "Catastrophic caldera-forming eruptions II: The subordinate role of magma buoyancy as an eruption trigger." Journal of Volcanology and Geothermal Research 305 (2015): 100-113

Urbani, S., V. Acocella, and E. Rivalta. "What drives the lateral versus vertical propagation of dikes? Insights from analogue models." Journal of Geophysical Research: Solid Earth 123, no. 5 (2018): 3680-3697.

Urbani, S., V. Acocella, and E. Rivalta. "What drives the lateral versus vertical propagation of dikes? Insights from analogue models." Journal of Geophysical Research: Solid Earth 123, no. 5 (2018): 3680-3697.

Thanks for pointing out these references, that we have read. But we have not added them to this manuscript, as we agree with the reviewer that this manuscript should not be the place to discuss the role of magma buoyancy in breaking crust.

Referee #3:

This is a novel and important study of patterns of unrest preceding the 2021 eruption at Fagradalsfjall, Iceland. The study makes a convincing case for a period of decreasing deformation following high rates of deformation and immediately preceding the eruption (see my comments on seismicity rates below), and for a model of tectonic strain release to explain the observed temporal patterns of unrest rates. The study provides strong evidence in support of an emerging paradigm (in contrast to the FFM paradigm) of short-term precursory quiescence before eruptions, and thus ultimately warrants publication in Nature. However, I

have several concerns regarding presentation and interpretation that should be addressed before the paper is acceptable for publication.

We thank the reviewer for the important comments, that we have all addressed as explained here below.

Major concerns:

1. The motivation for the study and its importance as given in the Introductory section is relatively weak, and could be greatly strengthened by enumerating additional prior examples of short-term precursory quiescence worldwide in a larger range of contexts (stratovolcanic eruptions (e.g., Pinatubo, Redoubt), phreatomagmatic eruptions (e.g., Suwanose-jima, Telica). As it stands only one other example, from Eyjafjall, Iceland, is mentioned in the introduction.

Motivation for the study and its importance has been strengthened, and more examples given of short-term precursory quiescence – in line with these suggestions. See new text at end of first paragraph in main text in the revised manuscript.

2. My major concern is that the purported decline in seismicity rate (as shown in Figure 1b, top panel) is not entirely convincing. There is a clear decline in the rate of large-magnitude earthquakes (and thus cumulative magnitude) as shown in the middle panel. However, the slope of the cumulative number of earthquakes curve is minimal at best, indicating the possibility that the large magnitude earthquakes and their aftershocks dominate the trend. This is a challenging issue to assess. Perhaps one means of addressing it would be to examine the rate of lower magnitude earthquakes through time (what is the magnitude of completeness for the catalog?). Or to attempt to deconvolve mainshock-aftershock sequences from the catalog by, e.g., removing the periods immediately following mainshocks from the plot.

We have reanalysed the seismicity rate in response to this comment, to better evaluate how seismicity rate evolves in comparison to the clear decline in the rate of large-magnitude earthquakes and the cumulative magnitude. In the end, we have, however, decided to not go into too much detail in the revised manuscript in this respect

For the period analyzed here, we estimate the magnitude of completeness (M_c) to be around M_1 , and therefore we limit our analysis to earthquakes $M > 1$ (see new text in Methods under routine earthquake locations).

We have divided up the presentation in the main text so we first cover the clear decline in earthquake magnitudes and seismic moment (and show that now in Fig 1 prior to the seismicity rate). We realized that a more complete description of the evolution of the seismicity was required in the paper to address the issue of the decline in seismicity rate. We have added an explanation of how activity is divided on the northern and southern dyke segments, and along the central axis of the plate boundary. After this we describe the seismicity rate evolution, that shows overall a decline only in the last few days prior to the eruption. This is further analysed by subareas in Figure 1.

3. (Lines 154-155 and Methods) - I do not think that a model of a cylindrical conduit is

appropriate, as the magma transport structure is clearly a fissure all along its depth.

Only a part below the dyke is modelled as a cylinder. We now justify our approach at the beginning of paragraph on conduit flow modelling in Methods. The approach is also discussed at the end of Methods. The wording in the main text has been modified.

4. (Statement beginning on L199). I strongly disagree with the characterization of the initial Fagradalsfjall seismicity as 'distal volcano-tectonic'. Distal volcano-tectonic earthquakes are earthquakes occurring in a volume of rock that is separated in distance from the intruding magma, which is not the case for the Fagradalsfjall earthquakes which all occur proximal to an intruding dike. The fact that the ultimate vent for the Fagradalsfjall eruption is some distance from the early (proximal to intruding dike) earthquakes does not meet the (arguably ambiguous) definition of 'distal VTs' given in ref 25 (from its source paper - White and McCausland 2016). Given the numerous issues with the published definition of the term 'distal VT' and their characterization and interpretation ((see discussion in Wauthier et al. 2016, EPSL; and Meyer et al. 2021, JVGR, for example), I recommend that this term and references to the White and McCausland studies be avoided entirely.

We have followed this advice. In light of this comment, we find more appropriate to compare our findings to other studies. See changes in the final paragraph of the manuscript.

5. I agree with the general interpretation (paraphrasing L208-211) that progressive lowering of crustal stress in rifting environments will exert a first-order control on temporal patterns of precursory seismic moment release and inflation. It would be useful to discuss this interpretation in the context of the large body of work on active vs passive rifting in East Africa (e.g., Ebinger et al. 2010; Keir et al. 2013; Pallister et al. 2010, and many others). For example, does this interpretation provide insight into long-term intrusive/eruptive ratios in rift settings, for example?

Our interpretation is now discussed in the context of observations at a number of locations around the world, including East Africa. See changes in the in the final paragraph of the manuscript.

We have not addressed the question if our interpretation provides insight into long-term intrusive/eruptive ratios in rift settings. The question might be best addressed with a study of the overall processes building up the full thickness of the crust, requiring a combination of studies of seismic crustal structure, tectonics, eruption and preferably studies of complete rifting episodes. The activity we describe is only one rifting event in what may be a longer episode of activity. We consider this topic more appropriate for a separate later study, so we have not addressed it here.

Minor comments/edits:

L80 - "eastward" -> "northeastward"?

Changed.

L89 - "N295E" - Legend for Figure 1b (lower panel) says "325 degrees"?

We have corrected a mistake and checked all displacement directions (in Fig. 1 and text) are correct.

L92-94 - I think the sentence beginning with "When" can be omitted.

This sentence has been omitted.

L94 - "It is also evident.." -> "It is also evident (from InSAR)..."

Changed.

L115-117 - It seems that the seismicity to W is early, and occurs before the south segment seismicity.

The description of the seismicity has been improved – see last paragraph of the chapter on “Pre-eruptive seismicity and deformation”.

L158-159 - The sentence as written is unclear - does this mean you're assuming that all of the sources (including dike) were active for a long time prior to eruption, or is stress accumulated over a long time? To clarify I suggest rewording as follows: "the strain and stress fields due to all the sources combine in a constructive manner to release stress accumulated by plate movements over a long time period prior to the eruption."

This sentence has been shortened and simplified in the revised version.

L413 - "47,000"

L414 - "7,500"

L416 - "Czech"

These corrections have been implemented.

L661 - "dominating fault structure" -> "dominant fault orientation and sense of slip"

Changed.

Fig 1a - The color bar dates are confusing (red is apparently ~2 months after eruption onset). Also meaning of "50 mm" is confusing.

Presentation in Fig. 1 has been improved. Hopefully it is clear now.

Fig 2 - Which pixels are used for calculations in F? (Same issue applies to EF 1)

The figure caption has been clarified.

Fig 3 - What is the meaning of the blue asterisk?

This is now explained in the figure caption.

Fig 3b - I suggest reversing the blue axis (so that 0 is on the bottom and 40 is on the top), as

the current presentation is non-intuitive.

We tried different versions of this figure, including the reversing only the blue axis. However, we find the presentation most intuitive if the two types of inferred parameters are shown in such a manner that the curves overlap (demonstrating a correlation). We modified the figure so now it has both axes reversed compared to the earlier version.

Reviewer Reports on the First Revision:

Referee #3:

I have read the revised version of the manuscript "Deformation and seismicity decline preceding a rift zone eruption at Fagradalsfjall, Iceland" by Sigmundsson et al. and find that all of my concerns have been addressed and that I can now recommend that the manuscript be accepted for publication. As noted in my original review, I feel that study provides strong evidence in support of an emerging paradigm of short-term precursory quiescence before eruptions, and thus represents an exciting and insightful advance. I congratulate the authors for this excellent work!

Best regards,
Diana Roman

Author Rebuttals to First Revision:

Reykjavik, Iceland, 23 June 2022

Regarding: Nature manuscript 2021-04-06983B

Response to reviewer comments.
(Comment from reviewer in italics, our response in normal font)

Referee #3:

I have read the revised version of the manuscript "Deformation and seismicity decline preceding a rift zone eruption at Fagradalsfjall, Iceland" by Sigmundsson et al. and find that all of my concerns have been addressed and that I can now recommend that the manuscript be accepted for publication. As noted in my original review, I feel that study provides strong evidence in support of an emerging paradigm of short-term precursory quiescence before eruptions, and thus represents an exciting and insightful advance. I congratulate the authors for this excellent work!

*Best regards,
Diana Roman*

******END******

The reviewer is happy with the changes we made in the initial revision of the paper, and no further modifications are requested.

There are only very minor changes made in the manuscript to improve clarity and correct a few small errors, and we have added 4 references (3 in the main text, and 1 in Methods). These are included to put our analyses into better perspective. These small changes do not alter the content of the manuscript in any way.

On behalf of the authors.

Freysteinn Sigmundsson